# Climate-driven deoxygenation elevates fishing vulnerability for the ocean's widest ranging shark

**Marisa Vedor[1,2], Nuno Queiroz[1,3]†\*, Gonzalo Mucientes[1,4], Ana Couto[1], Ivo da Costa[1], António dos Santos[1], Frederic Vandeperre[5,6,7], Jorge Fontes[5,7], Pedro Afonso[5,7], Rui Rosa[2], Nicolas E Humphries[3], David W Sims[3,8,9]†\***

[1]CIBIO/InBIO, Universidade do Porto, Campus Agrário de Vairão, Vairão, Portugal; [2]MARE, Laboratório Marítimo da Guia, Faculdade de Ciências da Universidade de Lisboa, Av. Nossa Senhora do Cabo, Cascais, Portugal; [3]Marine Biological Association of the United Kingdom, The Laboratory, Citadel Hill, Plymouth, United Kingdom; [4]Instituto de Investigaciones Marinas, Consejo Superior de Investigaciones Científicas (IIM-CSIC), Vigo, Spain; [5]IMAR – Institute of Marine Research, Departamento de Oceanografia e Pescas, Universidade dos Açores, Horta, Portugal; [6]MARE – Marine and Environmental Sciences Centre, Faculdade de Ciências da Universidade de Lisboa, Lisbon, Portugal; [7]Okeanos - Departamento de Oceanografia e Pescas, Universidade dos Açores, Horta, Portugal; [8]Centre for Biological Sciences, Highfield Campus, University of Southampton, Southampton, United Kingdom; [9]Ocean and Earth Science, National Oceanography Centre Southampton, Waterfront Campus, University of Southampton, Southampton, United Kingdom

**\*For correspondence:**
nuno.queiroz@cibio.up.pt (NQ);
dws@mba.ac.uk (DWS)

†These authors contributed equally to this work

**Competing interests:** The authors declare that no competing interests exist.

**Abstract** Climate-driven expansions of ocean hypoxic zones are predicted to concentrate pelagic fish in oxygenated surface layers, but how expanding hypoxia and fisheries will interact to affect threatened pelagic sharks remains unknown. Here, analysis of satellite-tracked blue sharks and environmental modelling in the eastern tropical Atlantic oxygen minimum zone (OMZ) shows shark maximum dive depths decreased due to combined effects of decreasing dissolved oxygen (DO) at depth, high sea surface temperatures, and increased surface-layer net primary production. Multiple factors associated with climate-driven deoxygenation contributed to blue shark vertical habitat compression, potentially increasing their vulnerability to surface fisheries. Greater intensity of longline fishing effort occurred above the OMZ compared to adjacent waters. Higher shark catches were associated with strong DO gradients, suggesting potential aggregation along suitable DO gradients contributed to habitat compression and higher fishing-induced mortality. Fisheries controls to counteract deoxygenation effects on shark catches will be needed as oceans continue warming.

## Introduction

Dissolved oxygen (DO) content of the global ocean is declining (ocean deoxygenation) due to sea temperature warming, increased stratification, and changing circulation, and the interactions of these processes with hypoxia-inducing biological activity (**Keeling et al., 2010**; **Gilly et al., 2013**; **Watson, 2016**; **Schmidtko et al., 2017**; **Breitburg et al., 2018**; **Levin, 2018**; **Laffoley and Baxter, 2019**; **Stramma and Schmidtko, 2019**). Permanent oxygen minimum zones (OMZs) are present in all the world's ocean basins and occur where DO reaches low (hypoxic) levels of $<0.45$–$1.00$ ml $O_2$

$l^{-1}$ in the depth range ~200–800 m (**Keeling et al., 2010**; **Gilly et al., 2013**; for oxygen units used see Materials and methods). Ocean deoxygenation has resulted in horizontal and vertical expansions of OMZs with the prospect of profound effects on marine biota because long-term DO declines are particularly acute in OMZs (**Stramma et al., 2008**; **Gilly et al., 2013**; **Breitburg et al., 2018**).

Levels of DO have a major role in structuring marine ecosystems (**Grantham et al., 2004**; **Ekau et al., 2010**; **Gilly et al., 2013**; **Levin and Le Bris, 2015**; **Breitburg et al., 2018**; **Penn et al., 2018**; **Laffoley and Baxter, 2019**) because the vast majority of organisms have a low hypoxic threshold, where concentrations of ~1.35–2.70 ml $O_2$ $l^{-1}$ cause hypoxic stress (**Vaquer-Sunyer and Duarte, 2008**; **Ekau et al., 2010**). OMZ vertical expansions may alter microbial processes key to nutrient cycling and gas fluxes (**Levin and Le Bris, 2015**), change predator–prey dynamics (**Ekau et al., 2010**; **Stewart et al., 2013**), and shift distributions, abundance, and capture risk of ecologically and commercially important species (**Ekau et al., 2010**; **Gilly et al., 2013**; **Prince and Goodyear, 2006**; **Prince et al., 2010**; **Stramma et al., 2012**; **Deutsch et al., 2015**). In particular, OMZ expansion may have significant implications for top predator ecology and fisheries (**Gilly et al., 2013**; **Stramma et al., 2012**) since shoaling hypoxic layers are expected to compress the habitat of pelagic fishes (**Prince and Goodyear, 2006**; **Prince et al., 2010**; **Stramma et al., 2012**; **Leung et al., 2019**). Hypoxia-based habitat compression is the habitat loss associated with expanding hypoxia, signified by the shoaling of cold, hypoxic water reducing the depth of the oxygenated surface mixed layer (**Stramma et al., 2012**). For large tropical pelagic fishes such as tunas and billfishes, water layers with DO concentrations 3.0–3.5 ml $O_2$ $l^{-1}$ have been recognised as a generalised lower habitat boundary associated with waters overlying OMZs that limits depth distributions (**Lowe et al., 2000**; **Ekau et al., 2010**; **Stramma et al., 2012**). This predicts shifts in predatory fish distributions even with mild hypoxia occurring in waters above OMZs (**Vaquer-Sunyer and Duarte, 2008**; **Ekau et al., 2010**; **Sims, 2019**). Nonetheless, the increased susceptibility of large pelagic fishes to fisheries due to shoaling OMZs has been hypothesised (**Gilly et al., 2013**; **Prince and Goodyear, 2006**; **Prince et al., 2010**; **Stramma et al., 2012**) but has not been quantified directly.

Pelagic sharks are ecologically important oceanic apex predators (**Hammerschlag et al., 2019**) with comparatively high physiological demands for oxygen (**Payne et al., 2015**) that have been widely observed in the surface layers above OMZs (e.g. **Vetter, 2008**; **Nasby-Lucas et al., 2009**; **Abascal et al., 2011**; **Queiroz et al., 2016**; **Queiroz et al., 2019**; **Sims, 2019**). This raises an important question: will expanding volumes of oxygen-depleted water, such as expanding OMZs, reduce the vertical extent of pelagic sharks into preferred but contracting surface habitat, increasing their risk of capture by surface fisheries? Tens of millions of pelagic sharks are caught each year by largely unregulated fisheries in the high seas (areas beyond national jurisdiction, ABNJ), leading to overfishing and declining catch rates documented for many species (**Baum et al., 2003**; **Ferretti et al., 2010**; **Worm et al., 2013**; **Campana, 2016**; **International Commission for the Conservation of Atlantic Tunas, 2017a**; **International Commission for the Conservation of Atlantic Tunas, 2019**). Despite high fishing pressure, there has been limited management of commercially important pelagic sharks where OMZs occur in ABNJ (**Campana, 2016**; **Queiroz et al., 2019**), such as for blue *Prionace glauca* and shortfin mako *Isurus oxyrinchus* sharks that are the most common sharks caught by Atlantic pelagic fisheries (**Queiroz et al., 2016**; **Oliver et al., 2015**; **Sims et al., 2018**). This suggests pelagic sharks including blue shark could be at growing risk from deoxygenation through reduction of the habitable space available and, therefore, potentially higher capture rates in those affected areas that overlap with pelagic fisheries. However, no studies have tested the hypothesis of hypoxia-based habitat compression in pelagic sharks acting to increase the capture risk of sharks to surface fisheries above OMZ areas, or considered specifically how catch rates are affected by shoaling hypoxia.

The eastern tropical Atlantic (ETA) OMZ provides an appropriate system to study in this context because expansion of this OMZ has been recorded over the last 50 years, with DO declines there being particularly intense compared to other OMZs (**Stramma et al., 2008**). For example, in this OMZ the vertical extent with DO concentrations <2.0 ml $O_2$ $l^{-1}$ – levels known to affect pelagic fish physiology and behaviour (**Ekau et al., 2010**; **Sims, 2019**) – increased in thickness by 85% between 1960 and 2006 (**Stramma et al., 2008**), with more recent observations of exceptionally low DO concentrations of <0.05 ml $O_2$ $l^{-1}$ ('dead zones') in some ETA mesoscale eddies (**Karstensen et al., 2015**). Therefore, hypoxic water of 1.5–3.5 ml $O_2$ $l^{-1}$ becoming increasingly prevalent in the pelagic zones above the core ETA OMZ (with concentrations <0.9 ml $l^{-1}$; **Karstensen et al., 2015**) as it

expands are likely to induce threshold physiological and behavioural responses of pelagic fishes leading to abundance and distributional changes (*Lowe et al., 2000*; *Ekau et al., 2010*), including those of sharks (*Sims, 2019*). Despite this, no targeted studies have investigated how shoaling hypoxia influences pelagic shark behaviour and susceptibility to fisheries.

We investigated these unknowns by satellite tracking blue sharks, the world's widest ranging shark, and recording horizontal and vertical habitat use with data-logging tags in waters above the ETA OMZ and adjacent areas. The outer boundaries of OMZs have strong gradients in DO concentration, whereby moving a short horizontal distance means the mixed layer depth (MLD) characterising the boundary between upper oxygenated and deeper hypoxic water rises upwards steeply (large DO gradient) (*Schmidtko et al., 2017*). These hypoxia depth gradients mimic the process of OMZ expansion through a space-for-time approach. Therefore, blue sharks were tracked and diving behaviour recorded in these areas to enable the effect of OMZ 'expansion' on shark dive patterns to be examined. We developed models to determine environmental drivers of vertical habitat use of sharks by including surface and at-depth DO, water temperature, salinity, and three biological productivity proxies in environmental models to estimate how shark maximum daily dive (MDD) depth was affected in both OMZ and normally oxygenated adjacent areas.

We also sought to understand from empirical observations how pelagic fishing effort and blue shark catches by longlines were distributed across the ETA OMZ area in relation to hypoxic water. The blue shark is highly migratory and the pelagic shark most frequently caught by pelagic longline fisheries worldwide and including in the Atlantic Ocean (*Oliver et al., 2015*; *Queiroz et al., 2016*). Classified by the International Union for the Conservation of Nature (IUCN) Red List as Near Threatened globally, this species has few catch restrictions anywhere in the world. Catch limits for Atlantic blue sharks were, however, agreed in November 2019 indicating concerns that the Atlantic populations are at potential risk of overexploitation (*International Commission for the Conservation of Atlantic Tunas, 2019*). Therefore, we tracked fishing patterns of the entire Spanish and Portuguese longline fleets – two of the most important oceanic fleets in the Atlantic that retain sharks (*Queiroz et al., 2019*) [deploying 15% of all hooks *International Commission for the Conservation of Atlantic Tunas, 2017a*] – in relation to the ETA OMZ and adjacent areas to quantify fine-scale differences in fishing effort and intensity directly. Finally, spatially referenced catch data for blue shark from Spanish longline fleet logbooks were used to examine how retained catches (hooking mortality) were distributed across the ETA OMZ area.

## Results

### Shark movements near hypoxic zones

A total of 55 adult or sub-adult blue sharks were tracked in the North Atlantic with either an Argos (position only) satellite transmitter or a satellite-linked depth and temperature-recording tag (pop-off satellite archival transmitter, PSAT) (*Figure 1*; *Supplementary file 1*). One shark, S1, was double-tagged. The purpose of using Argos transmitters was to determine the broad extent of horizontal movements and space utilisation of sharks in OMZ and adjacent areas of the North Atlantic, and as a spatial context for assessing potential habitat selection of sharks in the OMZ area, while PSAT tags were used to record spatially explicit swimming depths of sharks directly in relation to modelled DO and other environmental variables in OMZ and adjacent areas. Of the 22 sharks tagged with Argos transmitters between 2009 and 2014 in fully oxygenated waters adjacent to the ETA OMZ, 10 sharks displayed sustained southern movements of which six utilised waters above the OMZ (*Figure 1A*). To test whether sharks showed habitat selection for waters above the OMZ area when encountered we compared the movements of individuals to correlated random walk models with move steps (*Jaine et al., 2014*) and bearings randomly and independently drawn from each respective real shark trajectory (Materials and methods) (*Supplementary file 2*; *Figure 1—figure supplement 1*). Five (out of six) sharks spent significantly more time above the ETA OMZ than respective random models (*Supplementary file 2*). To investigate more directly whether waters above the ETA OMZ may be a preferred habitat we tagged 10 blue sharks with Argos transmitters above the OMZ in 2017 (*Figure 1A*; *Supplementary file 1*) and recorded movements post-tagging. Three sharks (S24, S25, and S27) stayed entirely within the region above the OMZ for the tracking durations (22–40 days), and while seven eventually left the area, two of the seven sharks remained for a significantly longer

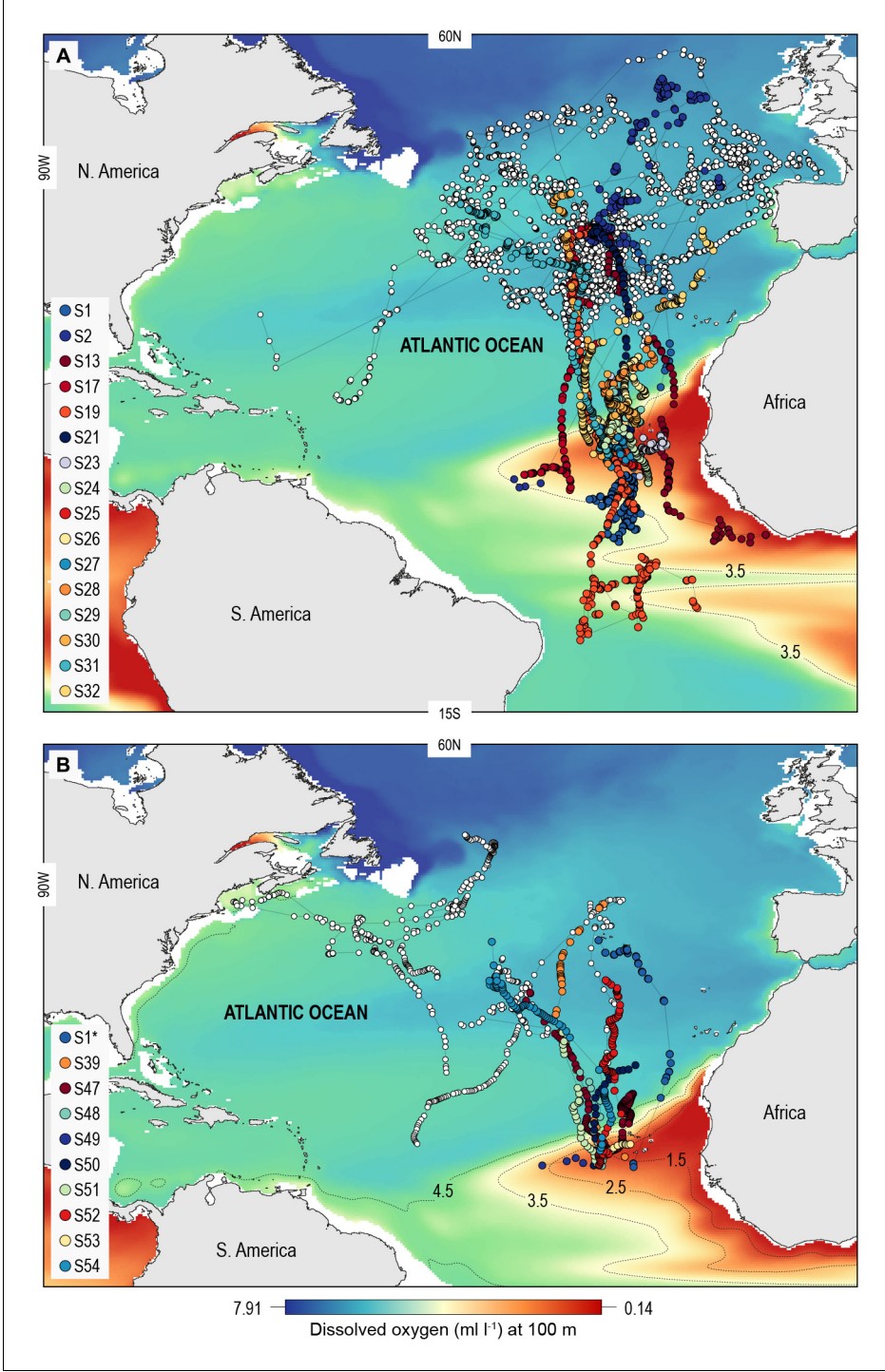

**Figure 1.** North Atlantic-wide movements of blue sharks in relation to the eastern tropical Atlantic (ETA) oxygen minimum zone (OMZ). Sharks were tracked with (A) ARGOS satellite transmitters (deployed in the Azores and above the ETA OMZ) and (B) pop-off satellite-linked archival transmitters (PSAT) (deployed in west and central North Atlantic, and above the ETA OMZ) overlaid on modelled dissolved oxygen concentrations (at 100 m). Coloured trajectories indicate individual sharks that moved into waters above the OMZ, which is illustrated using different oxyclines (1.5–4.5 ml $O_2$ $l^{-1}$ at 100 m depth). Note that shark S1* was double-tagged and appears in both panels.

The online version of this article includes the following figure supplement(s) for figure 1:

*Figure 1 continued on next page*

*Figure 1 continued*

**Figure supplement 1.** Assessing blue shark habitat preference for surface waters above the oxygen minimum zone (OMZ).

period than what would be expected by random movements prior to leaving (S23 and S29; tracked for 73 and 125 days, respectively).

## Shark diving patterns

Of the 16 sharks tagged with PSATs outside the OMZ area in 2010–2011, six made southerly movements from more oxygenated waters with three individuals (S1, S39, and S47) encountering waters above the OMZ (*Figure 1B*; *Table 1*). A further seven sharks (S48–54) were PSAT-tracked in waters above the ETA OMZ in 2017 (*Figure 1B*; *Table 1*). A vertical shift to shallower depths (increased shoaling) occurred as sharks moved across the core OMZ area compared to dive depths in adjacent (normoxic) waters (*Figure 2*). For example, the MDD depths of shark 47 in adjacent waters to the OMZ ranged between 500 and 1400 m, whereas above the core OMZ area MDD depths were 200–520 m (*Figure 2A*; *Table 1*). Similarly, the deepest MDD depths of shark 52 reached 1480 m in adjacent areas, but were mainly shallower than 250 m in the OMZ (97.6% of time) and only once reached 980 m (*Figure 2B*; *Table 1*). This pattern of shallow MDD depths in the ETA OMZ was also evident for sharks 53 and 54 (*Figure 2C and D*).

Overall, the mean MDD depth reached by individuals that encountered the OMZ was 754 m compared to a mean depth of 1250 m outside the OMZ (*Table 1*). For all sharks, the mean number of dives >600 m was an order of magnitude lower in waters above the core OMZ than more oxygenated waters outside (mean deep dives per track: above the OMZ = 1.1; adjacent to OMZ area = 10.8; *Table 1*; *Figures 2* and *3*). In addition, it was evident from dive data that although blue sharks undertook dives in the OMZ area (down to 1552 m) potentially taking them through hypoxic water (*Figure 3B and D*), these were of relatively short duration given that close to 100% of time was spent <500 m (*Figures 3A and C* and *4*), with 91% of time in depths <250 m (*Table 1*).

Changes in diving behaviour were concomitant with reductions in estimated concentrations of modelled DO and the calculated partial pressure of oxygen ($pO_2$) in the vicinity of each shark (after taking into account changing temperature and salinity with depth; see Materials and methods). Sharks tagged outside the OMZ area compared to inside potentially experienced an average

**Table 1.** Summary of vertical movement data for pop-off satellite archival transmitter (PSAT)-tracked blue sharks that encountered waters above the oxygen minimum zone (OMZ) area.
DNR denotes a tag that did not report.

| Shark ID | % Time upper 250 m | | Number dives below 600 m | | Maximum dive depth | |
|---|---|---|---|---|---|---|
| | OMZ | Outside | OMZ | Outside | OMZ | Outside |
| S1 | 59.7 | 69.9 | 2 | 6 | 680 | 696 |
| S39 | 98.5 | 83.8 | 0 | 19 | 288 | 1464 |
| S47 | 90.7 | 74.0 | 0 | 29 | 520 | 1400 |
| S48 | 88.8 | 92.9 | 2 | 7 | 1174 | 1390 |
| S49 | 98.6 | - | 1 | - | 930 | - |
| S50 | DNR | 85.7 | 1 | 2 | 804 | 1463 |
| S51 | 88.4 | 87.2 | 0 | 2 | 336 | 712 |
| S52 | 97.6 | 89.2 | 2 | 24 | 984 | 1480 |
| S53 | 98.0 | 84.7 | 3 | 1 | 1552 | 1264 |
| S54 | 97.9 | 92.1 | 0 | 7 | 272 | 1384 |
| *Mean* | *90.9* | *84.4* | *1.1* | *10.8* | *754.0* | *1250.3* |
| *S.D.* | *12.5* | *7.8* | *1.1* | *10.5* | *419.5* | *316.4* |

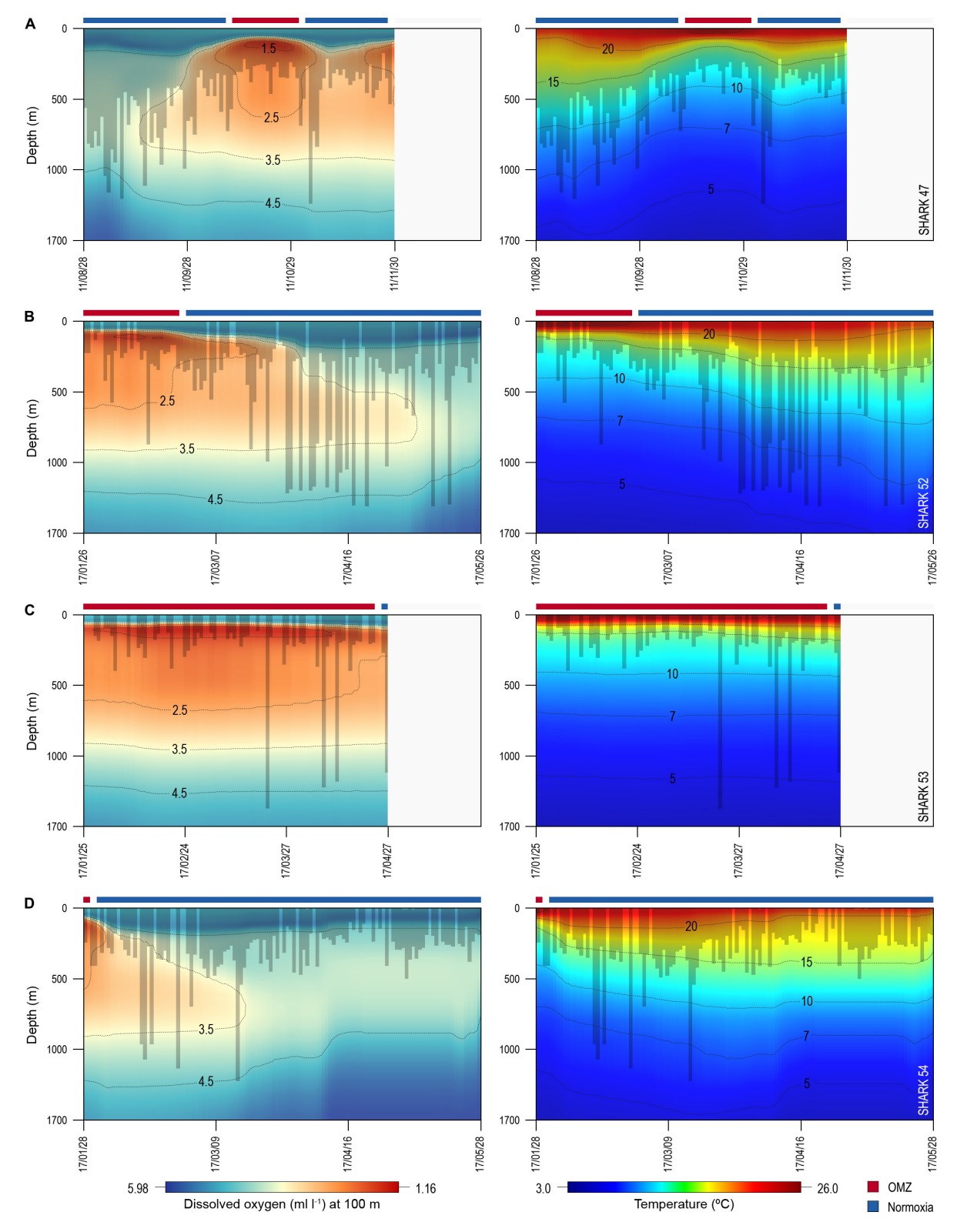

**Figure 2.** Vertical movements of blue sharks in relation to dissolved oxygen (DO) concentration and water temperature. Daily minimum-maximum depth plots (shaded area) for sharks 47 and 52–54, overlaid on DO concentration data (left panels) and water temperature (right panels) from the surface to 1700 m. Red and blue bars at the top of each panel denote the period spent inside (red) and outside (blue) the area overlying the oxygen minimum zone (OMZ). Numbers in the right panels denote oxyclines (ml O$_2$ l$^{-1}$) and in the left panels isotherms (°C).

*Figure 2 continued on next page*

*Figure 2 continued*

The online version of this article includes the following figure supplement(s) for figure 2:

**Figure supplement 1.** Vertical movements of blue sharks in relation to environmental variables.

decrease from 4.23 (±0.82 S.D.; minimum of 2.00 ml l$^{-1}$, maximum of 5.66 ml l$^{-1}$) to 2.48 ml O$_2$ l$^{-1}$ (±1.18 S.D.; min. of 1.17 ml l$^{-1}$, max. of 5.17 ml l$^{-1}$), corresponding to a decrease in the mean pO$_2$ from 13.39 (±3.08 S.D.; min. of 6.23 kPa, max. of 18.33 kPa) to 8.01 (±4.64 S.D.) kPa, with a minimum of 3.58 kPa and a maximum of 18.37 kPa (*Figure 2—figure supplement 1*). PSAT-tracked sharks in 2017 experienced a similar transition from 2.71 ml O$_2$ l$^{-1}$ (±1.15 S.D.; min. of 1.38 ml l$^{-1}$; max. of 5.11 ml l$^{-1}$) within the area above the OMZ to 4.10 (±0.88 S.D.; min. of 1.97 ml l$^{-1}$; max. of 5.49 ml l$^{-1}$) in more oxygenated waters outside the OMZ (*Figures 3* and *4*). This corresponded to a mean pO$_2$ of 8.70 (±4.55 S.D.) and a minimum of 4.27 kPa (max. of 18.48 kPa) above the OMZ compared with a mean pO$_2$ of 13.31 kPa (±3.58 S.D.; min. of 6.14, max. of 18.33) in normoxia (*Figure 4* and *Figure 2—figure supplement 1*).

## Environmental influences on diving

Almost 100% of shark dive time in the OMZ was spent at depths <500 m even though the sea surface temperatures (SSTs) there were warmer (23–26°C) compared to dive locations outside the OMZ (*Figure 3A and B*). The warmer SST above the OMZ corresponds with the shallower MLD above the OMZ compared to adjacent waters (*Figure 2*). However, the temperatures encountered vertically by sharks down to MDD depths were similar between the two areas (mean ± S.D. OMZ: 15.84 ± 5.18°C; normoxia: 15.75 ± 4.53°C), down to a minimum of ~5°C (maximum of 26.04°C) in the OMZ and a minimum of ~4°C (maximum of 26.84°C) in normoxia, albeit that these lower temperatures were encountered at shallower depths in the OMZ than outside (*Figure 3D*).

Sharks entered low temperature waters (<7°C) during deeper dives outside the OMZ but such waters were not generally reached when in the OMZ area, even though the 10°C isotherm was shallower in the OMZ (*Figure 2*). Shark 47, for example, regularly reached depths with water temperatures <7°C when moving near the edges of the OMZ and in adjacent waters, but did not enter <10°C water when in the core OMZ except on 1 day (*Figure 2A*, right panel). Shark 52 showed similar

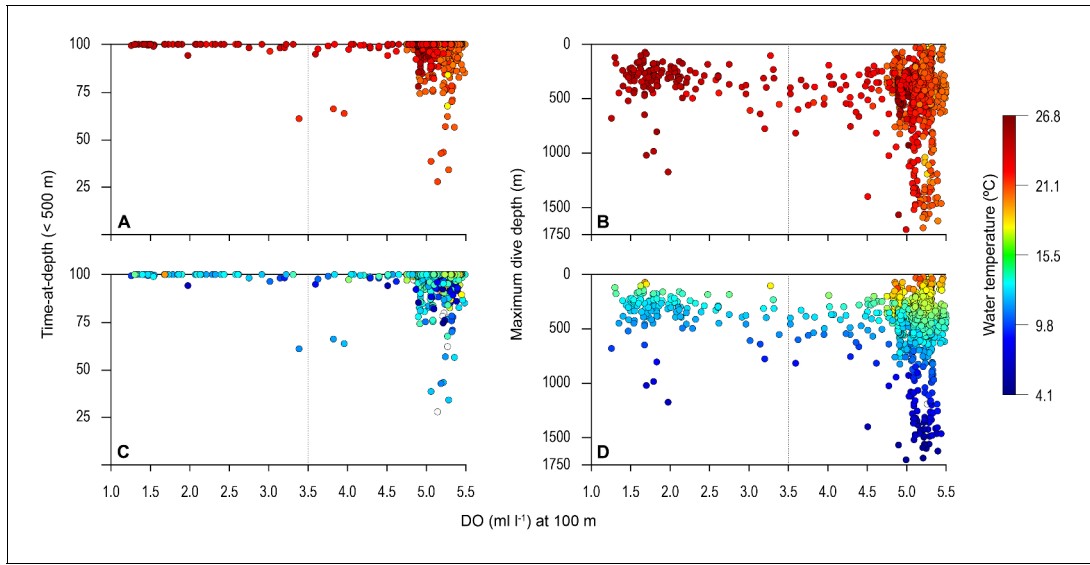

**Figure 3.** Vertical distribution of blue shark movements above oxygen minimum zone (OMZ) and adjacent waters. The percentage of tag-recorded time spent by blue sharks per day in the upper 500 m of the water (A and C) and maximum daily dive (MDD) depths (B and D) in relation to the dissolved oxygen (DO) concentration at 100 m depth. Time-at-depth (TAD) and MDD depth are given in relation to sea surface temperature (SST) (A and B) and temperature at depth (C and D). Each marker plotted summarises percentage TAD for 1 day. The dashed vertical line in each panel denotes the lower DO concentration indicated by the generalised additive mixed model (GAMM) below which MDD depth begins to decrease (3.5 ml O$_2$ l$^{-1}$).

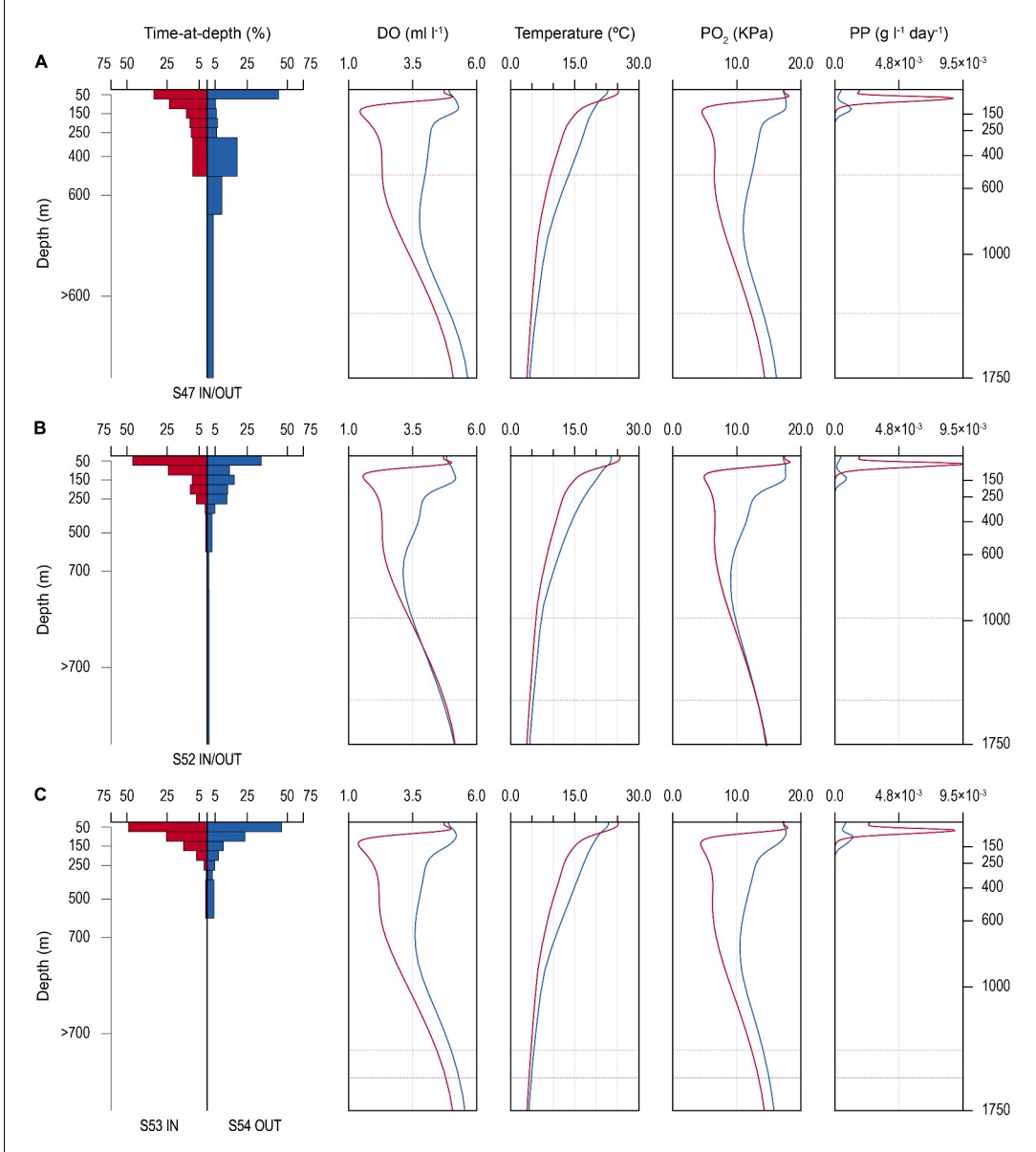

**Figure 4.** Time-at-depth of blue sharks in oxygen minimum zone (OMZ) and adjacent areas in relation to environmental variables. Vertical depth use ratio (left panel) between waters above the OMZ (red) and adjacent waters outside the area of OMZ (blue) for sharks 47 and 52 (**A and B**); depth use ratio comparing depth distributions of shark 53 above the OMZ with shark 54 outside the area above the OMZ (**C**). In A–C, the panels on the right show the average dissolved oxygen (DO) concentration, water temperature, pO₂, and net primary production (NPP) concentration between the surface and 1750 m, and horizontal dotted lines denote the MDD depth of each shark recorded inside (red) and outside (blue) the OMZ area.

responses, with substantially fewer deep dives into <10°C waters when in the OMZ compared to adjacent areas (*Figure 2B*, right panel). This indicates for these sharks at least that they were capable of tolerating <7°C waters at depth outside of core OMZs. However, when the 7°C isotherm was shallower in the OMZ area they did not reach those depths, demonstrating that cold water did not limit MDD depth in the same manner across OMZ and adjacent habitats.

The vertical gradients in water temperature where the sharks were recorded diving inside and outside the OMZ area were different principally in there being a sharper thermocline between 50 and 150 m above the OMZ compared to outside (*Figure 4*). This was also the depth range over

which DO and pO$_2$ decreased the most and which signalled the MLD characterised by oxygenated water above and more hypoxic water below. However, the temperature gradient (ΔT ˚C) from the middle of the MLD at 100 m to the MDD depth between waters inside the OMZ area and outside was 11.0˚C and 15.0˚C, respectively, for shark 47 (e.g. *Figure 4A*, compare red and blue lines in second panel from right) and 6.0˚C and 7.5˚C, respectively, for shark 52 (*Figure 4B*). Therefore, the ΔT values encountered by sharks during dives were similar inside and outside the OMZ area, although the minimum and maximum absolute temperatures were different (as mentioned above).

Similarity in thermal gradients encountered during diving by sharks was also evident from calculating the percentage time that individual sharks spent at different depths, DO concentrations, and water temperatures (*Figure 5*). For example, shark 47 spent most time at 0–50 m depth in water temperatures predominantly between 15˚C and 18˚C both inside and outside the OMZ area, and only outside the OMZ was appreciable time spent at around 400 m depth (*Figure 5A–C* left panels). In contrast, the time spent at different DO concentrations changed markedly over the same time course of diving movements, with most time in water of 4.8 ml O$_2$ l$^{-1}$ before entering the OMZ, compared to most time spent in water of 2.5 ml l$^{-1}$ inside the OMZ area despite the majority of water temperatures encountered remaining between 15˚C and 18˚C (*Figure 5B*, left panel). Results from shark 54 outside the ETA OMZ area confirms the predominance of surface orientated behaviour (0–50 m) and time spent diving to deeper depths (200–500 m) in waters with higher DO (5.5 ml O$_2$ l$^{-1}$) and a fairly constant temperature range (18–24˚C) (*Figure 5A–C*, right panels). Data from two further sharks (52 and 53) show similar patterns to sharks 47 and 54 both inside and outside the OMZ area (*Figure 5—figure supplement 1*).

Net primary production (NPP) was higher inside the OMZ area than outside, confirming the known relationship of higher phytoplankton growth in the ETA OMZ area as a consequence of increased stratification and warmer SST above the MLD, and which coincided with greater time spent by blue sharks in the uppermost 150 m water layer (*Figure 4*). Peak values of NPP occurred above the MLD at ~100 m depth inside the OMZ, values that were some nine times higher than peak values outside the OMZ in adjacent waters. NPP fell to very low concentrations below 200 m and remained low throughout deeper depths both inside and outside the OMZ.

## Modelling shark responses to environmental variables

The biotic and abiotic environmental variables influencing shark diving depths were likely to be complex in the OMZ region with strong horizontal and vertical gradients in DO, temperature, and biological productivity for instance. To explore this, we used a generalised additive mixed model (GAMM) to predict the MDD depth of tracked blue sharks as a function of surface and at-depth salinity, DO, water temperature, chlorophyll '*a*' and phytoplankton concentration, and NPP (Materials and methods). Results showed that MDD depth was best explained by a combination of DO at depth and SST and NPP at depth (*Table 2*). The variables of water temperature at depth, surface salinity and at depth, chlorophyll '*a*' and phytoplankton concentration (surface and at depth) did not contribute to the best fit model. Although the relationship between MDD depth and the selected variables in the best fit model was complex, MDD depth tended to decrease with decreasing DO at depth and SST, and increase with decreasing NPP at depth (*Figure 6*). The decrease in MDD depth was greatest where DO at depth decreased below 3.5 ml l$^{-1}$, whereas above this DO concentration MDD depth was not strongly affected (*Figure 6A*). At the highest levels of NPP at depth (100 m) MDD depth varied little, whereas when NPP decreased below ~0.0025 g l$^{-1}$ d$^{-1}$, shark MDD depth increased (*Figure 6B*). Increasing MDD depths occurred with increasing SST up to 24˚C; however, in waters with SST above 24˚C, which were common above the OMZ (*Figure 3B*), MDD depths decreased (*Figure 6C*).

We used the modelled relationships between shark MDD depth, DO, SST, and NPP to predict the extent of potential shark habitat compression occurring across the ETA OMZ region (*Figure 6D*; see Materials and methods). Predicted shallowest MDD depths were evident along the northern and western edges of the OMZ and off the continental shelf of western Africa (*Figure 6D*).

## Shark exposure risk to fisheries

To examine the potential increased susceptibility of sharks to fisheries in OMZ areas, we analysed the spatial overlap between the ETA OMZ and distributions of Spanish and Portuguese longline fleet

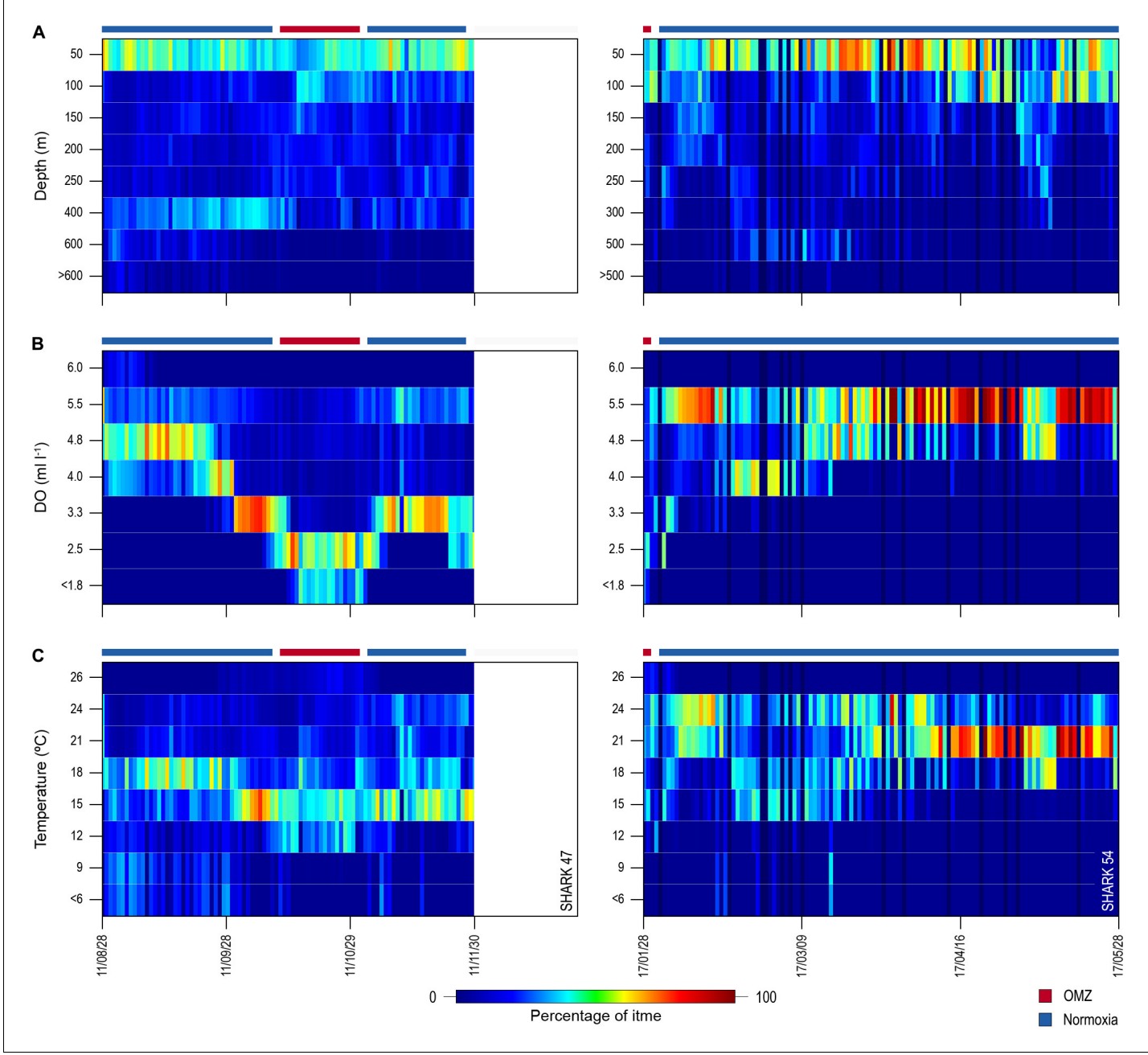

**Figure 5.** Changes in percentage time sharks spent at depth, dissolved oxygen (DO) concentrations, and water temperatures across oxygen minimum zone (OMZ) and adjacent areas. Left panels show percentage times of shark 47 and the right panels show shark 54.

The online version of this article includes the following figure supplement(s) for figure 5:

**Figure supplement 1.** Changes in percentage time sharks spent at depth, dissolved oxygen (DO) concentrations, and water temperatures across oxygen minimum zone (OMZ) and adjacent areas.

fishing locations (following the approach of *Queiroz et al., 2016*) (Materials and methods). We hypothesised that shallower MDD depths in the OMZ area compared to adjacent areas due to potential deoxygenation-driven habitat compression would act to increase the susceptibility of sharks to capture by surface longline fisheries. To explore this we mapped the extent of the ETA OMZ using the model-derived oxycline of 3.5 ml $O_2$ $l^{-1}$ at 100 m depth: this was chosen because the GAMM indicated that MDD depth begins to decrease below this DO concentration, inferring sharks show reduced preference for depths having these lower DO levels (*Figure 6A*). Analysis of

**Table 2.** Results of the GAMM model relating maximum daily dive (MDD) depth to environmental variables.

Std. error, standard error; $R^2$ (adj), adjusted r-squared for the model; s, smooth; edf, estimated degrees of freedom; Ref df, estimated residual degrees of freedom. SST, sea surface temperature; DO, dissolved oxygen concentration at depth; PP, net primary productivity at depth.

| | | Estimate | Std. error | T value | p-value |
|---|---|---|---|---|---|
| MDD depth (m) (n = 1085) | (intercept) | 347.58 | 14.39 | 24.16 | <0.001 |
| | | edf | Ref. df | F | p-value |
| $R^2$(adj) = 0.081 | s(DO) | 2.24 | 2.24 | 8.668 | <0.001 |
| | s(PP) | 2.183 | 2.183 | 10.16 | <0.001 |
| | s(SST) | 3.895 | 3.895 | 4.144 | 0.002 |

satellite-tracked fishing vessel movements in relation to this horizontal map of the subsurface OMZ extent showed different fine-scale fishing effort patterns (gear deployment locations) that reflected the different geographic regions and oceanographic regimes exploited. At the ocean-basin scale, fishing effort (active fishing days with gear deployed) was higher in the whole North Atlantic when compared to the South Atlantic, with longline fishing vessels spending on average 65 (±74 S.D.) and 30 (±30) fishing effort days per 1 × 1° grid cell, respectively (*Figure 7*; *Supplementary file 3*). At the regional scale in the North Atlantic, fishing effort and fishing intensity (proportion of total active fishing locations occurring in spatial clusters) were both higher above OMZ areas compared to adjacent waters outside (Mann–Whitney rank sum test, p<0.001; *Table 3*) (Materials and methods). Equally, in

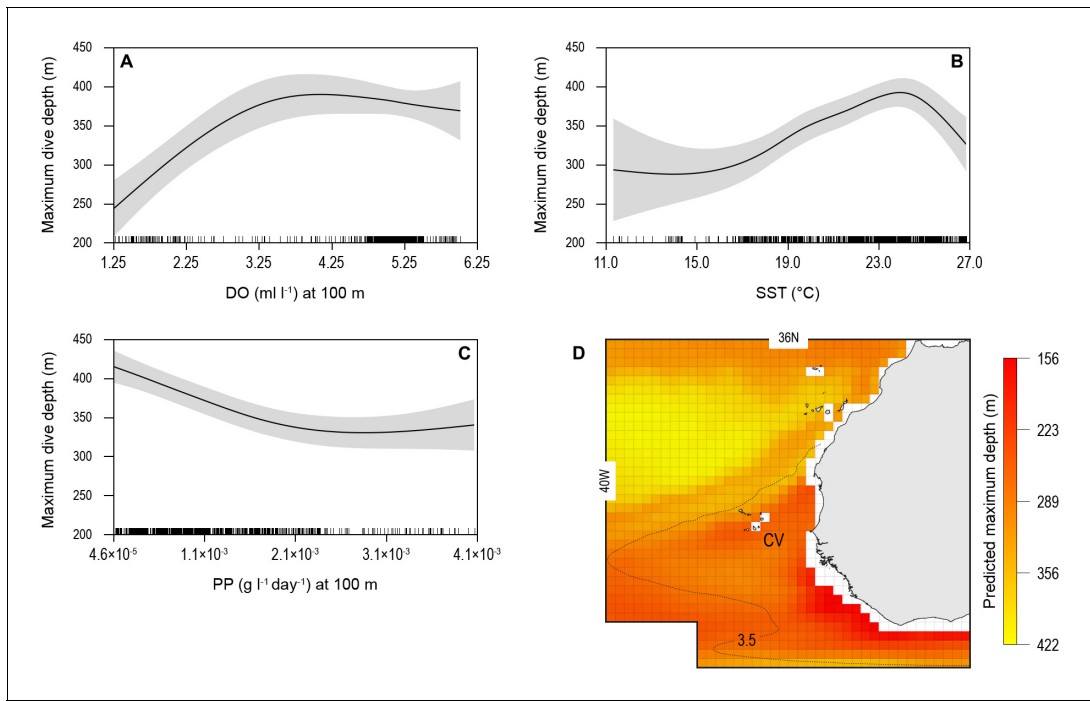

**Figure 6.** Generalised additive mixed model (GAMM) model relationships between blue shark dive depths and environmental variables with predicted present-day maximum daily dive (MDD) depths in the eastern tropical Atlantic (ETA) oxygen minimum zone (OMZ). Predicted response of modelled shark MDD depth (m) to (**A**) dissolved oxygen (DO) at depth, (**B**) sea surface temperature (SST), and (**C**) net primary productivity at depth (NPP). Modelled shark MDD depth increases with increasing DO at depth, with a lower threshold DO concentration at about 3.5 ml $O_2$ $l^{-1}$ below which MDD starts decreasing (**A**), and with increasing SST up to ca. 24°C, above which MDD depths decrease (**B**). The model showed an inverse relationship of MDD depth with NPP at depth, where modelled MDD depth declines with increasing NPP at depth (**C**). Continuous lines represent mean modelled MDD depth and shaded areas represent the standard error. (**D**) The predicted present-day extent of blue shark MDD depths off western Africa determined with the GAMM model relationships. Black dotted line denotes position of the oxycline of 3.5 ml $O_2$ $l^{-1}$ at 100 m depth.

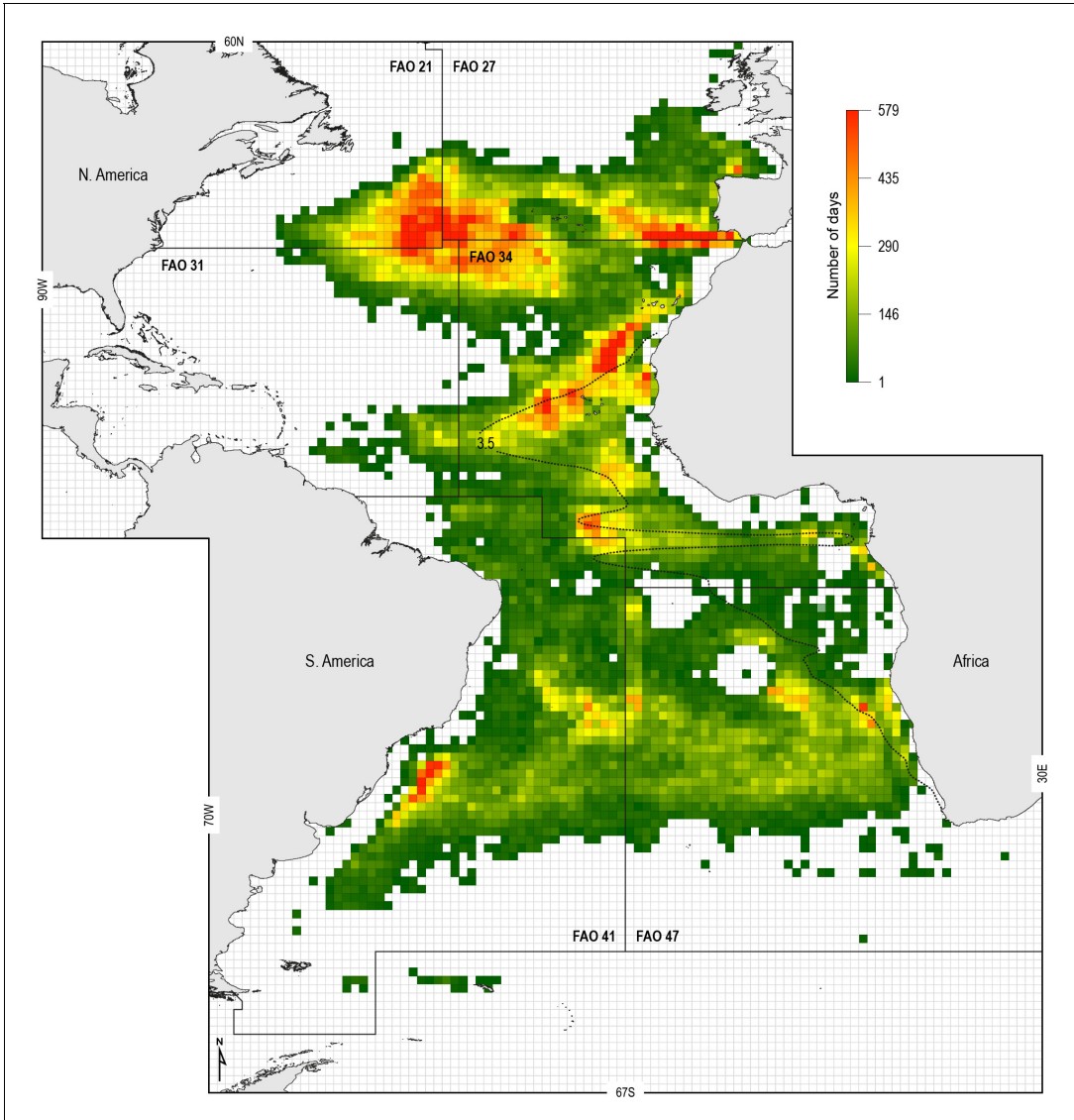

**Figure 7.** Spatial distribution of fishing effort of pelagic longline vessel fleets. Spanish and Portuguese longline fishing vessels (*n* = 322) were GPS satellite-tracked between 2003 and 2011 with the vessel monitoring system (VMS). Black dotted line: oxycline (3.5 ml O$_2$ l$^{-1}$ at 100 m) used to denote the positions of the North and South Atlantic hypoxic zones that we define by the lower habitat boundary (hypoxia threshold) for blue shark estimated by our generalised additive mixed model (GAMM) (see *Figure 6*) for blue shark maximum daily dive (MDD) in relation to dissolved oxygen (DO) concentration. FAO boxes shown for reference provided arbitrary areas for use in vessel spatial analysis.

**Table 3.** Fishing activity information for major fishing areas in the Atlantic with permanent OMZs indicate greater fishing intensity in waters above oxygen minimum zones (OMZs).

| FAO fishing area | Hypoxic (OMZ) or normoxic (more oxygenated) | Mean number of fishing days (per grid cell) | Fishing intensity (%) |
|---|---|---|---|
| FAO 34 (N Atlantic) | OMZ area | 72.7 | 65.0 |
| | Normoxic area | 62.5 | 57.9 |
| FAO 47 (S Atlantic) | OMZ area | 20.2 | 61.6 |
| | Normoxic area | 31.3 | 56.7 |

the South Atlantic fishing intensity was highest above OMZs (*Figure 7*; *Table 3*). Furthermore, it was evident that distinct hotspots of fishing effort occurred along the northern, western, and southern boundaries of the OMZ as defined by the oxycline of 3.5 ml $O_2$ $l^{-1}$ at 100 m depth (*Figure 7*).

## Blue shark catches in OMZ area

The blue shark catches of Spanish longline vessels as reported in logbooks showed higher catch per unit effort (CPUE; mean CPUE: kilograms of blue shark caught and retained per grid cell per longline set [day]) in the OMZ area compared to adjacent waters further north (>22°N) which were not affected by hypoxic waters at depth (*Figure 8A*). A higher number of grid cells with the highest CPUE occurred along the northern boundary of the OMZ (between 15 and 22°N), where there are very strong DO gradients horizontally and vertically (*Figures 1*, *2*, and *8A and B*). CPUE hotspots were also evident on the western extent (9–15°N and 30–41°W) and southern boundaries (south of 9° N) (*Figure 8A and C–G*). Relatively high CPUE in the western OMZ area was spatially more extensive than along the northern boundary; however, generally where catches were made the CPUE was lower (e.g. compare *Figure 8B* with E; *Table 4*). Few peak CPUE catches were made in the southern area, but where they did occur, they were located along the southern boundary with more fully oxygenated waters (*Figure 8A*).

Examining the fishing-induced blue shark mortality along transects through the OMZ areas showed CPUE increased most where DO at 100 m was generally decreasing (*Figure 8B–G* and *Figure 8—figure supplement 1*). Along the northern boundary (20.5 and 15.5°N transects) CPUE peaked where DO concentration at 100 m decreased most markedly, from ~5 ml $O_2$ $l^{-1}$ in the west (25–35°W) down to ~1 ml $O_2$ $l^{-1}$ in the east (15.0–15.5°W). For example, along the 20.5°N transect the peak CPUE was estimated to occur where the DO concentration was 3.51 ml $O_2$ $l^{-1}$ (*Figure 9A*). In the western region the decrease in DO concentration was less abrupt than further north (from ~4 ml $O_2$ $l^{-1}$ at 45°W to ~1 ml $O_2$ $l^{-1}$ at 15.5°W), and catches of blue sharks were more consistent in terms of CPUE and more broadly distributed along transects than further north (*Figure 8C* and *Figure 8—figure supplement 1*). Peak CPUE in individual grid cells occurred along transects where DO concentrations ranged from 1.65 to 3.49 ml $O_2$ $l^{-1}$ (*Table 4*). For all CPUE and DO data across grid cells, we estimated that blue shark CPUE peaked along transects in the OMZ area where DO at 100 m was 2.96 ml $O_2$ $l^{-1}$ (*Figure 9B*). Overall, the results show peak catches of blue shark occurred where sharp gradients in DO were present horizontally and vertically due to shoaling hypoxic water associated with the OMZ.

## Discussion

In this study we use the combined approach of satellite archival telemetry of blue sharks, environmental modelling, and spatial analysis of longline fishing effort and catch data to investigate shark responses to OMZ habitats and to examine vulnerability to capture by fisheries in deoxygenated regions. Overall, our results show that the ETA OMZ environment decreases the vertical range of sharks compared to adjacent waters consistent with the habitat compression hypothesis. Modelling showed MDD depth of blue sharks increased with increasing DO at depth and SST (up to 24°C) and decreasing NPP at depth. Interestingly, we found that water temperature at depth did not contribute to the best fit model, indicating that low DO at depth, specific SST values, and high NPP at depths within the uppermost 150 m layer were more important in reducing tracked blue shark MDD depth than gradients in temperature, salinity, and phytoplankton proxies in the water column of the OMZ. Furthermore, we found that CPUE of blue sharks by longline fishing vessels was higher in the OMZ area, primarily within fishing hotspots along the northern and western edges of the OMZ where the intensity of fishing was significantly higher and where predicted blue shark MDD depths were generally shallower.

### Shark diving behaviour

The blue shark is probably the ocean's widest ranging shark with documented individual movements of thousands of kilometres across entire ocean basins and from the surface down to over 1600 m depth (*Vandeperre et al., 2014*; *Queiroz et al., 2017*; *Queiroz et al., 2019*). Blue sharks in the course of these extensive movements encounter and appear capable of tolerating a very broad range of environmental conditions, such as water temperatures between 4°C and 30°C (*Howey et al., 2017*). Despite this, we found that the average maximum dive depth of tracked blue

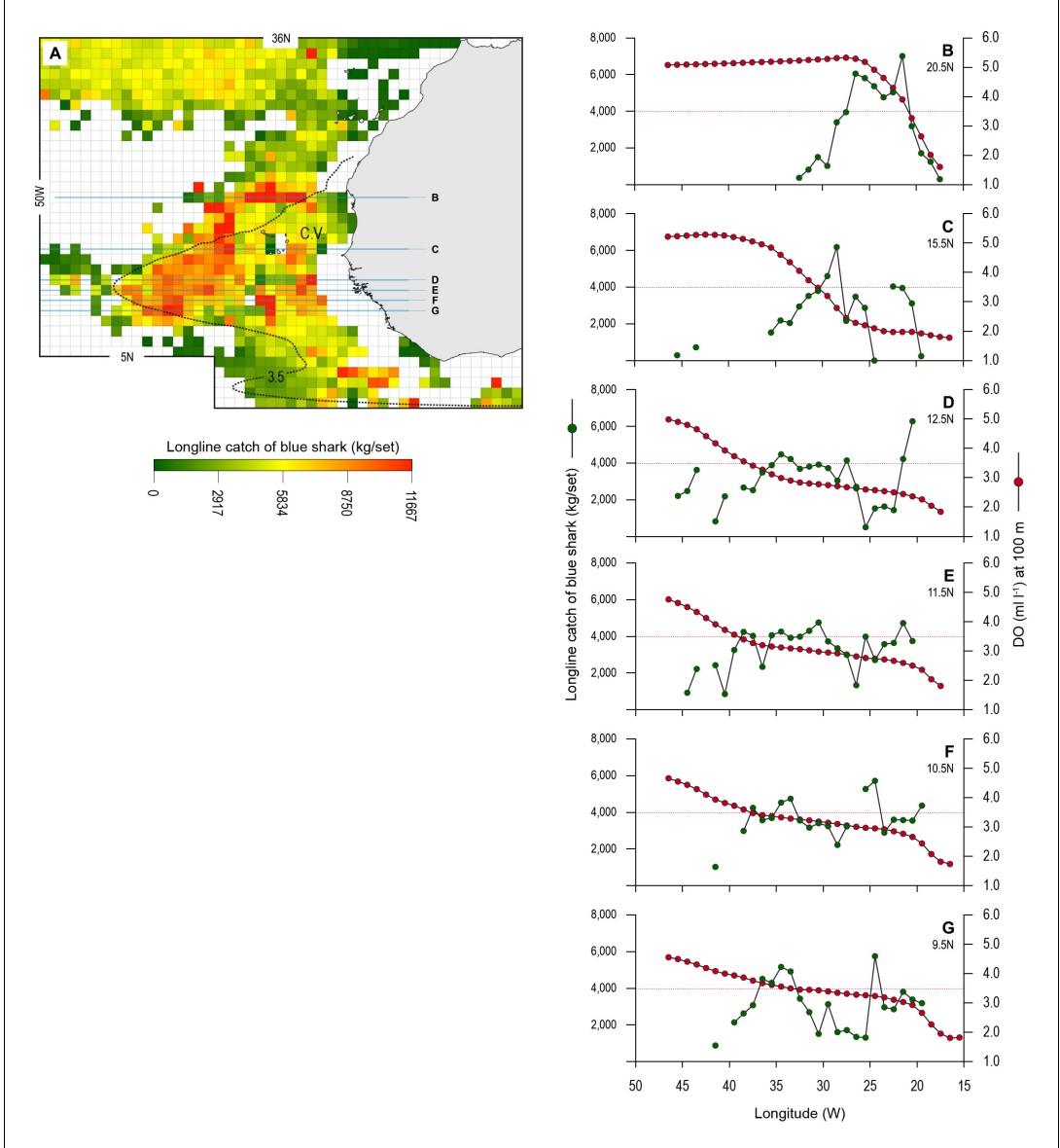

**Figure 8.** Longline catches of blue shark associated with the eastern tropical Atlantic (ETA) oxygen minimum zone (OMZ) and adjacent waters. (**A**) The mean catch per unit effort (CPUE) of blue sharks by Spanish pelagic longline vessels in the area of the OMZ (2013–2018). Horizontal lines denote the six data transects shown in **B–G**. The dotted line denotes the 3.5 ml $O_2$ $l^{-1}$ oxycline at 100 m depth representing the lower habitat boundary (hypoxia threshold) for blue shark estimated by our generalised additive mixed model (GAMM) (see *Figure 6*). (**B–G**) panels show how mean CPUE changes along each transect in relation to dissolved oxygen (DO) concentration at 100 m depth. The horizontal dotted line in each panel denotes the 3.5 ml $O_2$ $l^{-1}$ oxycline.

The online version of this article includes the following figure supplement(s) for figure 8:

**Figure supplement 1.** Cumulative mean catch per unit effort (CPUE) along each transect in relation to dissolved oxygen (DO) concentration at 100 m.

sharks in the ETA OMZ was 40% less than the mean depth attained outside the area, together with a greatly reduced frequency of deep diving below 600 m inside the OMZ. Tracked blue sharks were limited to shallower dives in the OMZ area compared to adjacent, normally oxygenated waters. In contrast, there was only a relatively small increase (6%) in the time spent in the upper 250 m layer when sharks were inside the OMZ area compared to outside it. This small difference in overall depth

**Table 4.** Summary of blue shark longline catch data across the eastern tropical Atlantic (ETA) oxygen minimum zone (OMZ). Longline catch of blue sharks were taken from Spanish pelagic longline vessel logbooks and aggregated into 1 × 1° grid cells. Each data transect analysed extended from 15.5°W to 46.5°W.

| Transect latitude (°N) | Number of longline sets | Total blue shark catch in all transect grid cells (tonnes) | Peak catch per unit effort (CPUE) (kg/grid cell/day) | Dissolved oxygen (ml $O_2$ $l^{-1}$ at 100 m) at peak CPUE |
|---|---|---|---|---|
| 20.5 | 665 | 3148 | 7025.02 | 3.49 |
| 15.5 | 1045 | 4021 | 6195.48 | 2.16 |
| 12.5 | 399 | 1436 | 6294.53 | 1.65 |
| 11.5 | 506 | 2006 | 4778.71 | 2.39 |
| 10.5 | 477 | 1815 | 5736.87 | 2.36 |
| 9.5 | 377 | 1386 | 5769.6 | 2.70 |

use reflects the preference of individual blue sharks to remain within the upper layer from where they undertake dives into deeper waters for relatively short periods of time. This depth use pattern is one of the common dive profiles documented for blue sharks in several previous studies conducted in different oceans, and is characterised by more time spent in the warmer upper layers during the night and more time at moderate depths during the day interspersed with much deeper dives into cooler waters (*Stevens et al., 2010*; *Queiroz et al., 2012*; *Howey et al., 2017*; *Queiroz et al., 2017*). Limitation of the vertical extent of deep dives in OMZ areas compared to adjacent areas was also apparent for white sharks (*Carcharodon carcharias*; *Nasby-Lucas et al., 2009*) and shortfin mako sharks (*Isurus oxyrinchus*; *Abascal et al., 2011*) along the northern and southern edges of the eastern Pacific OMZ, respectively. Similarly, our results show blue shark vertical movements in the OMZ area principally comprised a reduced mean maximum depth and frequency of deep dives >600 m, rather than a substantial shift in the temporal use of the uppermost water layer.

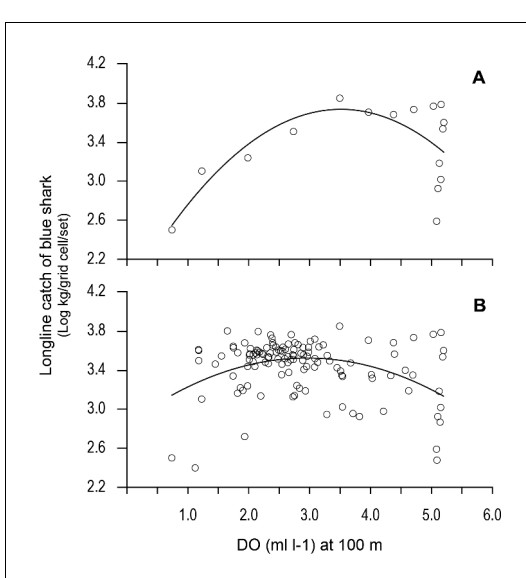

**Figure 9.** Variation in longline catches of blue shark along transects as a function of dissolved oxygen (DO) at depth. Catch per unit effort of blue shark (mean CPUE: kilograms of blue shark/grid cell/fishing set) along (**A**) the 20.5°N transect of the northern boundary of the eastern tropical Atlantic (ETA) oxygen minimum zone (OMZ) and (**B**) CPUE for all grid cells along all six transects in relation to modelled DO (at 100 m depth). Solid lines are quadratic fits to the data to provide an estimate of the DO concentration at 100 m depth supporting an estimated peak catch of blue shark. Equations and statistics of the fitted quadratic models: (**A**) $\log_{10}CPUE = 1.833 + 1.082\,DO - 0.1539\,DO2$ ($n = 16$, adj. $r^2 = 0.37$, $F = 5.49$, p=0.019) and (**B**) $\log_{10}CPUE = 2.846 + 0.4570\,DO - 0.07732\,DO2$ ($n = 123$, adj. $r^2 = 0.14$, $F = 10.88$, p<0.0001).

## Environmental drivers of shark dive depths

The reduced maximum dive depths and the lower frequency of deep dives by blue sharks were driven by sharp horizontal and vertical gradients in physical and biological variables present in the OMZ. The GAMM identified the principal drivers of blue shark MDD depth to be DO at depth, SST and NPP at depth, with MDD depth increasing as DO at depth increased from 1.25 to 3.5 ml $O_2$ $l^{-1}$, as SST increased to 24°C (above which MDD decreased with increasing SST) and as NPP at depth decreased. These results support the hypothesis that blue shark vertical habitat became compressed as they moved from normally oxygenated waters into the ETA OMZ area, which has both low DO at depth, high SST, and high primary production above the MLD.

Our finding that DO at depth is a major driver of blue shark MDD depth in areas above OMZs

supports the hypoxia-driven habitat compression hypothesis but also identifies other contributing factors of SST and NPP at depth. Considering DO concentrations, the GAMM results indicate a lower DO habitat boundary – defined here as the DO concentration below which MDD depth becomes more limited by further decreases in oxygen content – that was estimated by our model to be ~3.5 ml $O_2$ $l^{-1}$ at ~18°C. There were similarities in the DO concentrations reducing blue shark MDD depth with those observed for other large pelagic fishes. The lower habitat boundary DO concentration we found for blue shark is consistent with studies showing depth distributions of yellowfin (*Thunnus albacares*) and skipjack (*Katsuwonus pelamis*) tunas are limited by reductions in oxygen content to only 3.5 ml $O_2$ $l^{-1}$ (*Gooding et al., 1981*; *Cayré and Marsac, 1993*; *Richard, 1994*; *Lowe et al., 2000*). Tracking-based or fishing depth-of-capture estimates of lower DO concentrations encountered by blue sharks do not identify strict habitat boundaries, however, since sharks may choose to enter lower DO concentration waters for a short time period – a behaviour confirmed by our tracking-based results – which may incur an oxygen debt. Rather, laboratory-based experimental whole-animal measurements of the critical oxygen level ($P_{crit}$) determine the level below which a stable rate of oxygen uptake (oxyregulation) can no longer be maintained and becomes dependent upon ambient oxygen availability (oxyconforming). $P_{crit}$ was found to average $5.15 \pm 2.21$ kPa for 151 fish species, ranging from 1.02 to 16.2 kPa, although only two species included were sharks (*Rogers et al., 2016*). For southern Bluefin tuna, *Thunnus maccoyii*, $P_{crit}$ was around 5.5 kPa at 19°C (*Rogers et al., 2016*); however, similar data for pelagic sharks are lacking. The mean $P_{crit}$ for the bottom-dwelling catshark *Scyliorhinus canicula* was determined to be 6.5 and 8.0 kPa at 12°C and 17°C respectively (*Rogers et al., 2016*), and between 10.5 and 11.9 kPa at 24–32°C for sandbar sharks, an obligate ram-ventilation species (*Crear et al., 2019*). For blue sharks, we estimated from modelled oxygen concentrations during vertical movements that the minimum oxygen conditions potentially encountered were 3.58–4.27 kPa at 12–18°C, suggesting blue sharks enter waters with low partial pressures of oxygen that in many fish species, including benthic sharks, would be oxygen levels that elicit oxyconforming responses (*Rogers et al., 2016*). Therefore, whilst blue sharks appear able to utilise low DO environments associated with the ETA OMZ for short time periods, our results overall indicate that the OMZ habitat reduced MDD depths and the frequency at which deep MDD depths were reached.

The DO concentrations that limit oceanic pelagic shark movements and behaviour in the wild are little known for the majority of species at present (*Ekau et al., 2010*; *Sims, 2019*). However, tracking-based estimates of potential hypoxic thresholds with DO concentrations of 1.2–3.5 ml $O_2$ $l^{-1}$ that we estimated for blue sharks also generally limit the dive depths of other pelagic sharks (*Vetter, 2008*; *Abascal et al., 2011*; *Nasby-Lucas et al., 2009*). For example, studies tracking shortfin mako (*I. oxyrinchus*) near the eastern tropical Pacific (ETP) OMZ indicated individuals generally remained in waters with >3 ml $O_2$ $l^{-1}$ and rarely encountered water with <2 ml $O_2$ $l^{-1}$ (*Vetter, 2008*; *Abascal et al., 2011*). Similarly, white sharks (*Carcharodon carcharias*) offshore in the ETP were typically associated with DO concentrations of >3 ml $O_2$ $l^{-1}$ (*Nasby-Lucas et al., 2009*), and salmon sharks (*Lamna ditropis*) in the eastern North Pacific were found to exploit low DO environments of <3 ml $O_2$ $l^{-1}$ (*Coffey et al., 2017*). Our study also showed blue sharks may occasionally enter low DO waters (<1.5 ml $O_2$ $l^{-1}$), which is proposed for other pelagic shark species (*Vetter, 2008*; *Abascal et al., 2011*; *Nasby-Lucas et al., 2009*). For example, the dive profile of a single scalloped hammerhead shark (*Sphyrna lewini*) in Baja California, eastern central Pacific, showed occupation of depths where modelled DO was estimated to be severely hypoxic (<0.5 ml $O_2$ $l^{-1}$) (*Jorgensen et al., 2009*). In addition, salmon sharks were tracked in waters where modelled-DO minimum concentrations ranged from 0.4 to 0.9 ml $O_2$ $l^{-1}$ (*Coffey et al., 2017*). The overall picture emerging from tracking sharks in relation to modelled DO, including this study, is that vertical extent is limited by low DO, with a general lower habitat boundary around 3.5 ml $O_2$ $l^{-1}$ but with shorter duration deeper dives into more hypoxic water possible.

## DO, temperature, and NPP effects

In previous studies of fish including pelagic sharks, the combination of water temperature and DO has been shown to an important constraint affecting vertical range (*Nasby-Lucas et al., 2009*; *Abascal et al., 2011*; *Coffey et al., 2017*). During vertical movements, individuals experience a wide range of available oxygen, combined with variations in temperature throughout the water column. These variations in oxygen and temperature affect blood–oxygen binding capacity, and

consequently, the tolerance of individuals to hypoxia (*Vaquer-Sunyer and Duarte, 2008*; *Deutsch et al., 2015*). Hypoxia tolerance represents the minimum environmental pO$_2$ ($P_{crit}$) with which an individual can regulate oxygen uptake (*Farrell and Richards, 2009*) and higher water temperatures decrease oxygen solubility, and consequently, available environmental pO$_2$ (*Tromans, 1998*; *Garcia, 2005*). In a recent study, the MDD depth of the salmon shark tracked with satellite archival transmitters in the northeast Pacific Ocean decreased when waters with low DO (<1–3 ml l$^{-1}$) and cold temperatures (<6℃) at depth were encountered, with both factors contributing to the best fit GAMM (*Coffey et al., 2017*). Although water temperature and DO can be important in determining shark dive depths, we found a less marked effect of temperature at depth with blue sharks diving into cold waters during deep dives both in the OMZ and adjacent waters, and the vertical change in temperature (ΔT ℃) below the MLD being similar in magnitude both inside and outside the OMZ. In contrast, we found, for example, most time was spent by a blue shark in low DO water of 2.5 ml O$_2$ l$^{-1}$ inside the OMZ area despite the majority of water temperatures encountered remaining between 15℃ and 18℃. The broad environmental tolerances of blue sharks likely explain why temperatures at depth appeared less important in determining MDD depth. The temperatures encountered by blue sharks in the OMZ (5–26℃) were within their normal, wide temperature range recorded in other studies (*Queiroz et al., 2012*; *Howey et al., 2017*), which suggests that the individuals we tracked were not solely limited in MDD depth by temperature at depth. We cannot discount the possibility that the combination of low DO and temperature at depth may affect blue shark MDD depth; however, we could not test this given that inside the OMZ the lowest DO concentrations encountered were coincident not with low temperatures but with a broad range of water temperatures from 10℃ to 20℃. This infers that DO at depth, SST, and NPP at depth were more important in driving MDD depth than colder temperatures at depth for blue sharks in the ETA OMZ.

SST has been shown in previous studies to be an important factor determining the habitat preferences of blue sharks in the North Atlantic (*Queiroz et al., 2016*), so it was expected that SST would contribute to the best fit model. The ETA OMZ is characterised by higher SSTs than adjacent waters (e.g. *Figure 2A*) and, indeed, we found that MDD depth increased with increasing SST up to 24℃, with SST >24℃ being observed along blue shark tracks across the core OMZ area that coincided with the shallowest MDD depths. The combination of high SST above the OMZ lowering oxygen solubility in water, together with the increased metabolic costs (oxygen consumption) of an ectothermic shark such as the blue shark associated with occupying elevated SSTs (*Payne et al., 2015*), may have acted to reduce the tolerance of blue sharks to undertake deeper dives into waters with low DO at depth.

The abundance and distribution of biological resources such as primary productivity has a central role in determining the habitats selected by consumers and predators in addition to abiotic factors. For example, the seasonal increase in phytoplankton abundance on the northeast-Atlantic continental shelf edge occurs a few weeks before an increase in zooplankton abundance and the occurrence of zooplankton-feeding basking sharks (*Cetorhinus maximus*) (*Sims et al., 2003*). Similarly, the MDD depths of blue sharks were also partly explained in our model by changes in NPP at depth, where NPP quantifies the rate at which phytoplankton produces biomass. We found that MDD depth decreased as NPP in the water column increased. At first this pattern seems counter-intuitive, since greater production at deeper depths might be expected to elicit deeper dives by a predator if prey species were strongly correlated with NPP. However, in the ETA OMZ area, the highest NPP concentrations occurred in the uppermost 150 m layer above the core OMZ. Therefore, our model results indicate that blue shark MDD depth decreased due to low DO at depth and also due to the influence of increasing phytoplankton biomass in the uppermost 150 m. The combined effects of low DO at depth and higher basal biomass in the upper layer above the core OMZ may lead to enhanced shark–prey interactions in the surface layers. Results support the hypothesis that some tracked blue sharks (10 of 16; 63%) exhibited selection of habitats above the core OMZ when encountered, which could be explained by enhanced foraging opportunities for sharks above and possibly within low DO environments of the OMZ that occurred in the uppermost 150 m layer. Blue shark feed on pelagic fish prey that may be habitat compressed into surface waters by the ETA OMZ, perhaps also due to the combined effects of low DO at depth and greater feeding opportunities in the upper layer. In addition, the diet of blue sharks in the North Atlantic is known to contain hypoxia-tolerant prey such as cephalopods (e.g. squid *Histioteuthis* spp.) that are present in deeper hypoxic waters

during the day and migrate vertically into more oxygenated, productive surface waters at night (*Biton-Porsmogeur, 2017*; *Childress and Seibel, 1998*). This prey migration into the upper layer above the core OMZ could be one factor driving the decrease in blue shark MDD depth in the OMZ area compared to adjacent waters that we observed.

Therefore, overall, our results support the hypothesis that expanding OMZs due to climate change (shoaling low DO; increasing SST and higher upper layer NPP) will cause habitat compression of blue sharks further into surface waters above expanding OMZs and reduce habitat volumes. The results strongly suggest that abiotic (low DO and high SST) and biotic factors (NPP, hence fish prey species) contribute to the observed decreased MDD depth of blue sharks in the ETA OMZ. This habitat compression suggests blue sharks will be increasingly at risk of capture by surface fisheries as a consequence of further deoxygenation of waters deeper than the surface mixed (oxygenated) layer overlying the OMZ. The habitat compression expected in surface waters due to climate-driven OMZ expansion and its concomitant effects on prey species distributions predicts that the potential susceptibility (availability and encounterability risks) of sharks to baited longline hook depths above OMZs are likely to be higher in the future.

A limitation of this study was that modelled DO data from oceanographic data sets were matched with locations along the paths of tracked pelagic sharks (*Sims, 2019*). Actual DO concentrations in the habitats selected by sharks may therefore differ from interpolated, modelled DO values. This is a limitation not only of the present study but indeed most previous shark studies. There is a need for direct DO measurements to assess more accurately how pelagic sharks' movements may be limited by low DO levels in the wild. Data-logging tags capable of measuring DO and swimming depths have been developed for recording DO levels that are directly encountered by free-ranging marine predators (*Bailleul et al., 2015*) including large sharks (*Coffey and Holland, 2015*). Oxygen saturations as low as 9.4% of normoxia ($\sim$0.8 ml $O_2$ $l^{-1}$ at 5°C) were recorded from tags on bluntnose sixgill sharks (*Hexanchus griseus*) off Hawaii, Pacific Ocean, during dives to nearly 700 m depth (*Coffey and Holland, 2015*; *Coffey et al., 2020*). DO sensor tag results to date confirm that very low DO concentrations are encountered and can be tolerated by large sharks during normal vertical movement. These types of sensors will enable new research on the physiological responses of ram-ventilating pelagic sharks to low DO in the wild (*Sims, 2019*) which, when combined with novel sea-going respirometers (*Payne et al., 2015*) for direct determination of $P_{crit}$, will pave the way for a mechanistic understanding of hypoxia tolerance in large shark species.

## Fishing effort and shark catches

The model output predicting how the OMZ environment affects blue shark vertical habitat suggests the susceptibility to capture by fisheries will be increased in areas where the DO shoals the most to further reduce vertical extent. This assertion was broadly supported in this study by the empirical fishing effort and catch data. This showed that fishing effort and blue shark CPUE were highest in the same discrete areas (hotspots) along the northern, western, and southern boundaries of the ETA OMZ, indicating both higher fishing intensity and capture success in those areas with greatest reduction in blue shark MDD depth. Strikingly, we found that blue shark catches peaked where horizontal DO gradients were sharpest, which suggests a profound effect of shoaling hypoxia on susceptibility of blue sharks to capture by pelagic longline vessels.

The results indicate longline fishing patterns above the ETA OMZ form among the most important fishing hotspots within the entire eastern Atlantic, particularly along the northern extent of the ETA OMZ. Fishing effort hotspots are formed where fishing effort is spatially concentrated. This suggests that oceanographic features associated with waters above the OMZ are a target area for surface longline vessels. Furthermore, higher fishing intensity in the OMZ area indicates that vessels spend less time searching for, or moving to, different fishing locations. As a result, vessels above the OMZ deploy longlines nearly continuously (daily) in localised areas rather than having to interrupt fishing by travelling for days between fishing hotspots (*Queiroz et al., 2016*). These areas of high fishing intensity are likely known to fishers from previous experience to be fishing hotspots supporting higher large fish catches (*Queiroz et al., 2016*). Although incomplete, longline fishery catch records for other species of pelagic sharks (e.g. *International Commission for the Conservation of Atlantic Tunas, 2017b*) and tunas (*Prince and Goodyear, 2006*; *Prince et al., 2010*) in surface waters above the ETA OMZ compared to adjacent areas support this conclusion.

Our results showed blue shark CPUE by Spanish longlines peaked where the modelled DO concentrations at 100 m were in the range 3.0–3.5 ml $O_2$ $l^{-1}$ (*Supplementary file 1*), which is a similar concentration range to a lower-habitat-boundary DO concentration of ~3.5 ml $O_2$ $l^{-1}$ that we determined from the GAMM (with tracking-derived MDD depth as the response variable). The similarity of these ranges suggests a potential mechanism underlying the peak catches observed. Here, blue sharks begin being limited in vertical extent by the 3.5 ml $l^{-1}$ DO surface (oxycline). Where there is a strong gradient between this preferred DO surface and deeper, less oxygenated waters, the movements of sharks along the more suitable oxygen gradients may therefore lead to aggregation along the boundary of the hypoxic zone. Previous studies on fish and invertebrates near low oxygen environments have reported higher mean catches, or steep increases in catches, relative to lower DO values (*Howell and Simpson, 1994*; *Song et al., 2009*; *Ekau et al., 2010*; *Craig, 2012*; *Potier et al., 2014*). For example, the effects of hypoxia avoidance by brown shrimp (*Farfantepanaeus aztecus*) and several demersal fish species in the Gulf of Mexico resulted in evading organisms aggregating short distances of 1–3 km just beyond the margins of the hypoxic zone and in a narrow region along the hypoxic edge (*Craig, 2012*). This suggested the potential for enhanced catches of those target and bycatch species along the hypoxic boundary (*Craig, 2012*). Likewise, several species of tunas frequently diving below the thermocline show a vertical compression to shallow depths in the presence of an OMZ (*Richard, 1994*; *Song et al., 2009*; *Potier et al., 2014*). This avoidance of hypoxic waters may be a behavioural mechanism to reduce physiological consequences of prolonged oxygen debt (*Richard, 1994*). Consequently, increased occurrence in well-oxygenated waters above OMZs increases the overlap between tuna vertical distribution and depths of gear deployments of surface longline fishing hooks and purse seiners (*Richard, 1994*; *Song et al., 2009*; *Potier et al., 2014*). The increased vulnerability of tunas to fisheries in OMZs is in line with the observed higher catches of tunas within these areas (*Song et al., 2009*; *Potier et al., 2014*). Similarly, the peak catches of blue sharks we observed in association with strong DO gradients may have resulted from movements limited by and, hence, aggregated along and above this boundary-edge oxycline, which may also have aggregated prey species, thus leading to higher use of these areas by blue sharks. The results highlight the potential for overexploitation of habitat-compressed pelagic sharks occurring along suitable oxygen gradients associated with expanding OMZs.

## Conclusions

Collectively, our findings suggest that blue sharks above the ETA OMZ will undergo habitat compression as they respond by avoiding lower DO concentrations and associating with high SST and NPP in the uppermost 100 m layer, and show preference for suitable DO gradients along the margins of OMZ areas – potentially due to the physiological limits of sharks (*Payne et al., 2015*; *Coffey et al., 2017*; *Sims, 2019*), to habitat compression of prey species (*Gilly et al., 2013*; *Childress and Seibel, 1998*), or even hypoxia-related visual impairment (*McCormick and Levin, 2017*). Our results propose that these responses combine to further increase shark risk of capture by surface fisheries as OMZs expand given that longline fisheries already target specific OMZ habitats where increased shark catches are made. The blue shark makes up ~90% of the total reported catch of pelagic sharks in the Atlantic (*Oliver et al., 2015*) and its fins are those most commonly traded in international markets (*Fields et al., 2018*). Reporting of catches remains poor, however (*Campana, 2016*), despite uncertainty about blue shark stock status (e.g. North Atlantic; *International Commission for the Conservation of Atlantic Tunas, 2017a*; *International Commission for the Conservation of Atlantic Tunas, 2019*). Our study predicts that blue shark habitat will not only become further compressed above OMZs in the future due to ongoing ocean deoxygenation (*Breitburg et al., 2018*), but, as a consequence, this species will become more susceptible to capture due to higher fishing intensity leading to higher CPUE than in more oxygenated adjacent areas. Management measures for threatened pelagic sharks, which specifically act to mitigate the effects of ocean deoxygenation on catch rates, may be required as oceans continue warming. Spatial management, such as high-seas large marine protected areas (LMPAs) around an OMZ for example, may be a management option necessary to consider in the warmer deoxygenating oceans of the future in addition to the need for more effective existing catch control measures to conserve shark populations.

## Materials and methods

### Shark tagging

A total of 55 blue sharks (*Prionace glauca*) were tagged in oceanic locations between 2009 and 2017 with 56 transmitters; one shark (S1) was double tagged (Argos and PSAT tags; *Supplementary file 1*). Sharks were captured on commercial style baited longlines deployed and, either (*i*) tagged at the surface off an auxiliary fibreglass boat (off the Azores) or (*ii*) brought alongside the vessel in the gear-hauling phase, lifted and tagged (at North Atlantic oceanic locations). Pop-off satellite-linked archival transmitter tags (PSATs) (Models PAT Mk10 and MiniPAT, Wildlife Computers, Redmond, WA, USA) were rigged with a monofilament tether covered with silicone tubing and looped through a small hole made in the base of the first dorsal fin. Depth, external temperature, and light-level parameters were sampled at varying intervals (from 1 to 10 s) and stored as summary data over set intervals of 6 hr, providing time-at-depth (TAD) and time-at-temperature histograms, as well as profiles of water temperature at depth. Argos-linked satellite tags (both Smart position-only tags – SPOT5, Wildlife Computers and KiwiSat K2F, Sirtrack Ltd., New Zealand) were attached to the first dorsal fin with nylon threaded rods or stainless-steel bolts, neoprene and steel washers, and steel screw-lock nuts. Tagging procedures were approved by the Marine Biological Association of the UK (MBA) Animal Welfare Ethical Review Body (AWERB) and licensed by the UK Home Office through Personal and Project Licences under the Animals (Scientific Procedures) Act 1986. Tagging procedures undertaken off the Azores were performed according to national Portuguese laws for the use of vertebrates in research, and the work and tagging protocol approved by the Azorean Directorate of Sea Affairs of the Azores Autonomous region (SRAM 20.23.02/Of.5322/2009), which oversees and issues permits for scientific activities.

### Track processing

The movement of PSAT-tagged sharks was estimated using either satellite relayed data from each tag or from archival data after the tags were physically recovered. Positions of each shark between attachment and tag pop-up were reconstructed using software provided by the manufacturer (WC-GPE, global position estimator program suite; Wildlife Computers, U.S.A.), where daily maximal rate-of-change in light intensity was used to estimate local time of midnight or midday for longitude calculations, and day-length estimation for determining latitude. Anomalous longitude estimates resulting from dive-induced shifts in the estimated timings of dawn and dusk from light curves were discarded from the data set using software provided by the manufacturer (WC-GPE). Latitude estimates were subsequently calculated for the previously obtained longitudes. An integrated state-space model (unscented Kalman filter – UKFSST [*Lam et al., 2008*] using spatially complete NOAA Optimum Interpolation Quarter Degree Daily SST Analysis data) was then applied to correct the raw geolocation estimates and obtain the most probable track. A regular time-series of locations was then estimated using a continuous-time correlated random walk Kalman filter, CTCRW (*Johnson et al., 2008*) performed in *R* (*crawl* package). Location class (LC) Z data (failed attempt at obtaining a position) were removed from the Argos-linked tag data set. The remaining raw position estimates (LC 3, 2, 1, 0, A, and B) were analysed point-to-point with a 3 m s$^{-1}$ speed filter to remove outlier locations. Subsequently, the CTCRW state-space model was applied to each individual track, producing a single position estimate per day. Argos positions were parameterised with the K error model parameters for longitude and latitude implemented in the *crawl* package (*Johnson et al., 2008*).

To obtain unbiased estimates of shark space use, gaps between consecutive dates in the raw tracking data were interpolated to one position per day. However, any tracks with gaps exceeding 20 days were split into segments prior to interpolation, thus avoiding the inclusion of unrepresentative location estimates (*Queiroz et al., 2016*; *Queiroz et al., 2019*).

### Oxygen concentration units

Fish tolerance to hypoxia is often measured in relation to the partial pressure of oxygen (pO$_2$) in the blood (*Farrell and Richards, 2009*; *Mandic et al., 2009*). Some authors highlight the importance of using pO$_2$ as a reference value for species tolerance to hypoxia rather than using an oxygen concentration in the water (*Mislan et al., 2016*). For conformity and ease of comparison with other

investigations of large pelagic predators and environmental oxygen completed to date (e.g. *Stramma et al., 2012*), and given that OMZs are oceanographic features defined by the concentration of oxygen dissolved in the water, we used ml $O_2$ $l^{-1}$ as the units of oxygen concentration in the water. Conversions follow those given in *Levin, 2018*. Nevertheless, we also analysed the diving movements of blue sharks in relation to partial pressures of oxygen calculated from modelled surface and at-depth DO, temperature, and salinity data and report these in the Results and discussion and in *Figure 4* and *Figure 2—figure supplement 1*. The partial pressure of oxygen in the water was calculated using R library '*rMR*', using temperature, salinity, and DO from CMEMS global ocean biogeochemistry non-assimilative hindcast (PICES; 1998–2016) and global ocean physics reanalysis (GLORYS12V1; 1993–2016) products, interpolated to matching depth levels for both products at 0.25° spatial resolution. Spatial geolocation error of shark tracking was taken into consideration by averaging $pO_2$ for 1.25° in latitude and 0.75° in longitude around each shark position.

## Environmental integration of movement data

Monthly modelled oxygen data (0.25° spatial resolution) were taken from the Copernicus Marine Environment Monitoring Service (CMEMS, http://marine.copernicus.eu/) global ocean biogeochemistry non-assimilative hindcast (PICES; 1998–2016) product for the Atlantic Ocean, ranging from 60°N to 70°S, 90°W to 30°E. DO concentration was extracted from the surface to 2000 m depth, and to account for the spatial error around real individual geolocations, oxygen data was averaged for 1.25° in latitude and 0.75° in longitude (5 × 3 pixel grid) around each shark position.

For the purpose of spatial analysis of blue shark horizontal movements, we defined the OMZ area in the ETA as the waters above the OMZ having DO concentrations below 3.5 ml $O_2$ $l^{-1}$ at depths shallower than 100 m (general depth of the thermocline in the ETA). This definition of the ETA OMZ area was adopted here because it was used in previous studies as the DO level known to induce stress in large tropical pelagic fishes (*Prince et al., 2010*; *Stramma et al., 2012*) and so was considered a reasonable proxy for the lower habitat boundary (hypoxia threshold) for blue sharks, an assumption confirmed by our GAMM analysis. As such, tracked sharks were considered to be inside the OMZ area when DO levels above 100 m decreased below 3.5 ml $O_2$ $l^{-1}$. TAD data were grouped into eight depth classes (0–50, 50–100, 100–150, 150–200, 200–250, 250–400, 400–600, >600 m; in 2017 the depth classes were: 0–50, 50–100, 100–150, 150–200, 200–250, 250–300, 300–500, 500–700, >700 m) that were calculated daily and aggregated into equal periods of time for sharks that were inside/outside the OMZ area. Dive data for each shark on the day prior to entering the OMZ area ('outside') were compared with data on the day following entry into the OMZ area ('inside').

## Correlated random walk simulations

To test the hypothesis that waters above the core OMZ may represent preferable foraging habitat for blue sharks, we used correlated random walks to compare the proportion of time (in days) spent in the OMZ region (when sharks were in an area with <3.5 ml $O_2$ $l^{-1}$ above 100 m) between tracked and model sharks (*Figure 1—figure supplement 1*). Randomised tracks had the same characteristics as the original tracks in terms of duration, length, and overall structure (or direction). Thus, each real shark track was randomised to be able to compare the original movement pattern to a random pattern having the same statistical distributions and overall structure (step lengths; path bearings). The track to be randomised was first analysed to determine the step lengths (distances between turns) and the bearing of each step. Each random track was started at the original start point (i.e. latitude and longitude) and subsequent, randomised locations were computed by drawing a step length and bearing randomly and independently from the real shark data. Using bearings, rather than turn angles, constrained the randomised track to an overall structure that more closely matched the original. Each randomised track had the same number of locations as the original track. For each tracked shark that encountered the OMZ, a total of 100 correlated random walks were simulated and a proportion test was performed (*Jaine et al., 2014*) to examine whether sharks remained above hypoxic regions longer than expected by chance (*Table 1*).

## Modelling shark behaviour and environment

A GAMM (*mgcv* package in *R*; *Wood, 2006*) was used to relate the MDD depth of tracked blue sharks to environmental variables. For each shark track day, maximum depths were retrieved from

PSAT depth/temperature data and the following variables were extracted from CMEMS global ocean biogeochemistry non-assimilative hindcast (PICES; 1998–2016) and global ocean physics reanalysis (GLORYS12V1; 1993–2016) products for both the surface and at depth (100 m) variables of salinity (PSU), DO concentration (ml $O_2$ $l^{-1}$), water temperature (°C), chlorophyll $a$ concentration (µmol $l^{-1}$), phytoplankton concentration (µmol $l^{-1}$), and net primary productivity (g $l^{-1}$ $d^{-1}$). Given the importance of $pO_2$ in hypoxia tolerance studies, initial models included $pO_2$ as an independent variable. However, the partial pressure of oxygen was calculated from modelled water temperature, salinity, and DO, which resulted in high collinearity between the variables and, ultimately, in models with low fit. As a result, this variable was removed from the model selection process.

Data exploration techniques were used to identify potential outliers and assess collinearity among independent variables. The top 10% of MDD depth data (i.e. dives deeper than 736 m) were considered outliers and discarded from further analysis. In addition, most of the extreme deep dives (89.8%) were performed outside the OMZ and were not therefore considered further in the MDD depth analysis. GAMMs were fitted assuming a normal distribution with identity link functions, as these outperformed models assuming a gamma distribution with identity and inverse link functions. Individual sharks were considered an independent sampling unit and were thus included as random effects. GAMMs were constructed by backward selection of individual smooth terms and previously chosen two-dimensional tensor product smooths to allow for testing of biologically meaningful interactions. The selected tensor product smooth terms were: (*i*) surface temperature × DO at depth; (*ii*) net primary productivity at depth × DO at depth; and (*iii*) surface temperature × primary productivity at depth. Model selection was based on Akaike Information Criterion (AIC), and at each stage of the selection process, selected smooth terms were replaced with linear parametric terms to reduce the chances of over-fitting, and nested models were additionally compared using significance testing (*Wood, 2006*). Smooth splines were selected with significance testing through comparison with the corresponding additive structures. Heteroscedasticity in the model residuals was accommodated by including a *varIdent* weighted variance structure (*Zuur et al., 2009*). Normal quantile–quantile plots of deviance residuals were assessed for normality of model residuals and fit. Homoscedasticity, model misspecification, and residual spatial auto-correlation were evaluated by inspecting plots of response residuals against fitted values, candidate explanatory variables, and spatial coordinates respectively. Spatial and temporal residual auto-correlation was further assessed by including respective covariate structures and comparing the nested models using significance testing (*Zuur et al., 2009*).

Modelled shark MDD depth in relation to present day SST, DO, and net primary productivity at depth in the OMZ region (FAO area 34) was used to predict shark MDD depths spatially. CMEMS environmental data between 2009 and 2016 was averaged to 1° spatial resolution. The fitted GAMM was then applied to 2009–2016 environmental conditions to predict the present shark MDD depth in 547 grid cells covering the OMZ in FAO area 34. The MLD above the thermocline is well oxygenated (close to 100% saturation for most oceanic regions) and the OMZ expansion is driven by the shoaling of the tropical and subtropical thermocline depth (*Schmidtko et al., 2017*). Therefore, the predicted MDD depth of blue sharks was limited to the maximum depth of the mixed layer. The predicted blue shark MDD depths from modelled relationships with DO, SST, and NPP were mapped to indicate potential areas where shark habitat compression may be occurring in the OMZ region.

## Vessel monitoring system data

Vessel monitoring system (VMS) data from 322 Spanish and Portuguese longliners (>15 m length) operating in the Atlantic Ocean from January 2003 to December 2011 were obtained from the respective official national fisheries monitoring centres. Each record contained the Global Positioning System (GPS) position of the vessel (accurate to <500 m), time stamp, and a vessel identification number. All records were anonymous with respect to the vessel registration number, dimensions, and administrative ports. To determine actual fishing locations where individual longlines were deployed, we used an algorithm (*Queiroz et al., 2016*) that detected sharp turning angles (>130°) that were considered to be the point between longline deployment and retrieval (see Figure 8—figure supplement 1 in *Queiroz et al., 2016*). Briefly, the algorithm proceeds by determining when a possible turn point is found, then the inbound leg is retraced until the distance travelled exceeds the longline length (between 80 and 100 km); the prior point is then taken as the start of deployment and the outbound leg is traced until the end-of-deployment point is determined in a similar fashion

(*Queiroz et al., 2016*). The algorithm undertakes a further check to determine if the endpoints are within a short distance of each other to confirm that a proper 'V' shape is defined (*Queiroz et al., 2016*). Algorithm-determined fishing locations were validated visually for individual vessel tracks selected at random and found to identify longline deployments accurately in all cases. Therefore, in this study, all movements between fishing locations were ignored (including trips to and from fishing ports), retaining only data pertaining to fishing activity (locations with gear deployed). To estimate fishing effort (number of days fishing with gear deployed), fishing data were first normalised by calculating daily centroids per individual vessel and then fishing days summed for each 1° × 1° grid cell. Fishing data were further analysed to identify areas of restricted search, or spatial clusters of longline deployment locations (termed fishing intensity) (*Queiroz et al., 2016*). Briefly, when searching for fish species, longliners move away from and back to the start position in three distinct phases: line deployment, soak time, and line retrieval. However, when sufficient numbers of target fish are found, the vessel remains stationary during soak time, thus allowing the longliners to target the same areas repeatedly, which results in (*i*) higher spatial concentration of active fishing locations (data points), (*ii*) a higher number of turn points per grid cell, and (*iii*) increased number of hours between turn points, which were identified by the filtering algorithm as area-restricted spatial (ARS) clusters. Fishing intensity was thus calculated as the number of fishing locations that were identified as ARS fishing activity divided by the total number of active fishing locations at a spatial resolution of 1°. Therefore, a fishing intensity value of 1 means that all active fishing locations were classified as ARS fishing locations and not as broad-scale searching locations (where fishing locations become linear across sets and not spatially clustered). Since both fishing effort and intensity were not normally distributed (Shapiro–Wilk test, $p < 0.05$), metrics were compared between major Atlantic Food and Agriculture Organisation of the United Nations (FAO) fishing areas using a Kruskal–Wallis one-way analysis of variance on ranks. Further analysis focused on major fishing areas that encompass permanent OMZs, namely FAO34 (eastern North Atlantic) and FAO47 (eastern South Atlantic). For both areas, monthly oxygen concentration data (at 100 m) was averaged for the time period considered in the study (2003–2011) and fishing metrics inside and outside the OMZs were compared using Mann–Whitney rank sum tests ($\alpha < 0.05$) since data was not normally distributed (Shapiro–Wilk test, $p < 0.05$).

## Analysis of blue shark catch data

Data on the magnitude and distribution of blue shark catches made in the Atlantic Ocean were obtained from the logbooks of Spanish pelagic longline vessels ($n = 133$) that recorded the spatially referenced biomass (kg) of blue sharks caught on each longline deployed (termed a longline set) (years: 2013–2018). Longlines were deployed at a depth of ~100 m in the OMZ area. Within each 1° × 1° grid cell we calculated the blue shark mean CPUE as the sum of the total biomass of blue shark caught within a grid cell divided by the sum of sets (days) of fishing effort in that grid cell. For these fleets, one longline set (~100 km long with ~1200 baited hooks) is deployed once per day, hence one set deployed is equivalent to 1 day of fishing effort. We related mean CPUE to the modelled DO concentration at 100 m (a proxy to indicate the degree of shoaling of the OMZ) within each grid cell along six data transects (at latitudes 20.5, 15.5, 12.5, 11.5, 10.5, and 9.5°N) spanning 15.5–46.5°W of the west African OMZ area and adjacent areas further west. These latitudes and longitudes were selected to encompass the range of CPUE values observed in the western African OMZ. To examine how the magnitude of blue shark CPUE changed with the shoaling OMZ (decreasing DO concentrations at 100 m) we calculated cumulative mean CPUE across each transect from west to east. To estimate the DO concentration at 100 m coincident with peak blue shark CPUE we fitted quadratic models to the distribution of $\log_{10}$(CPUE) (y axis) against DO at 100 m (ml $O_2$ $l^{-1}$) (x axis) for data along the 20.5°N transect, and for all transect data combined (Minitab 18, Minitab Inc).

## Acknowledgements

Funding was provided by the UK Natural Environment Research Council (NERC) (NE/R00997/X/1), European Research Council (ERC-AdG-2019 883583 OCEAN DEOXYFISH), NERC Oceans 2025 Strategic Programme (all to DWS), the Save Our Seas Foundation (DWS, NQ), Fundação para a Ciência e a Tecnologia (FCT) under PTDC/BIA/28855/2017 and COMPETE POCI-01–0145-FEDER-028855 (NQ, DWS), and MARINFO–NORTE-01–0145-FEDER-000031 (funded by Norte Portugal Regional

Operational Program [NORTE2020], under the PORTUGAL 2020 Partnership Agreement, through the European Regional Development Fund–ERDF), and Xunta de Galicia Spain under the Isabel Barreto Program 2009–2012 (GM). FCT supported NQ (CEECIND/02857/2018), MV (PTDC/BIA-COM/28855/2017), GM (PTDC/MAR-BIO/4458/2012), FV (CEECIND/03469/2017), and JF (through the strategic project UID/MAR/04292/2013 to MARE). Direcção Regional de Ciência e Tecnologia (DRCT) also supported JF (M3.1a/F/062/2016). DWS was supported by a Marine Biological Association Senior Research Fellowship. This research is part of the Global Shark Movement Project (globalsharkmovement.org).

## Additional information

### Funding

| Funder | Grant reference number | Author |
|---|---|---|
| Natural Environment Research Council | NE/R00997/X/1 | David W Sims |
| Fundacao para a Ciencia e a Tecnologia | CEECIND/02857/2018 | Nuno Queiroz |
| European Research Council | 883583 | David W Sims |
| Save Our Seas Foundation | 308 | Nuno Queiroz David W Sims |
| Fundação para a Ciência e a Tecnologia | PTDC/BIA/28855/2017 | Nuno Queiroz David W Sims |
| Fundação para a Ciência e a Tecnologia | COMPETE POCI-01–0145-FEDER028855 | Nuno Queiroz David W Sims |
| Norte Portugal Regional Operational Program | MARINFO–NORTE-01–0145-FEDER-000031 | Gonzalo Mucientes |
| Xunta de Galicia | sabel Barreto Program 2009–2012 | Gonzalo Mucientes |
| FCT | PTDC/BIA-COM/28855/2017 | Marisa Vedor |
| FCT | PTDC/MAR-BIO/4458/2012 | Gonzalo Mucientes |
| FCT | CEECIND/03469/2017 | Frederic Vandeperre |
| FCT | UID/MAR/04292/2013 | Jorge Fontes |
| DRCT | M3.1a/F/062/2016 | Jorge Fontes |
| Marine Biological Association | Senior Research Fellowship | David W Sims |

The funders had no role in study design, data collection and interpretation, or the decision to submit the work for publication.

### Author contributions

Marisa Vedor, Formal analysis, Validation, Investigation, Methodology, Writing - original draft, Writing - review and editing; Nuno Queiroz, Conceptualization, Data curation, Software, Formal analysis, Supervision, Funding acquisition, Validation, Investigation, Visualization, Methodology, Writing - original draft, Project administration, Writing - review and editing; Gonzalo Mucientes, Ana Couto, Investigation, Methodology, Writing - review and editing; Ivo da Costa, Software, Formal analysis, Validation, Methodology, Writing - review and editing; António dos Santos, Software, Supervision, Writing - review and editing; Frederic Vandeperre, Resources, Software, Formal analysis, Validation, Investigation, Writing - review and editing; Jorge Fontes, Resources, Formal analysis, Validation, Investigation, Methodology, Writing - review and editing; Pedro Afonso, Resources, Data curation, Supervision, Funding acquisition, Investigation, Methodology, Writing - review and editing; Rui Rosa, Resources, Supervision, Funding acquisition, Writing - review and editing; Nicolas E Humphries, Software, Formal analysis, Validation, Investigation, Methodology, Writing - review and editing; David W Sims, Conceptualization, Data curation, Formal analysis, Supervision, Funding acquisition, Validation,

Investigation, Methodology, Writing - original draft, Project administration, Writing - review and editing

### Author ORCIDs
Marisa Vedor (iD) https://orcid.org/0000-0001-7336-3732
Nuno Queiroz (iD) https://orcid.org/0000-0002-3860-7356
Gonzalo Mucientes (iD) http://orcid.org/0000-0001-6650-3020
Rui Rosa (iD) http://orcid.org/0000-0003-2801-5178
Nicolas E Humphries (iD) http://orcid.org/0000-0003-3741-1594
David W Sims (iD) https://orcid.org/0000-0002-0916-7363

### Ethics
Animal experimentation: Tagging procedures were approved by the Marine Biological Association of the UK (MBA) Animal Welfare Ethical Review Body (AWERB) and licensed by the UK Home Office through Personal and Project Licences under the Animals (Scientific Procedures) Act 1986. Tagging procedures undertaken off the Azores were performed according to national Portuguese laws for the use of vertebrates in research, and the work and tagging protocol approved by the Azorean Directorate of Sea Affairs of the Azores Autonomous region (SRAM 20.23.02/Of.5322/2009), which oversees and issues permits for scientific activities.

### Decision letter and Author response
Decision letter https://doi.org/10.7554/eLife.62508.sa1
Author response https://doi.org/10.7554/eLife.62508.sa2

## Additional files

### Supplementary files
• Supplementary file 1. Summary data for satellite-tagged blue sharks. F, female; M, male. Shaded rows indicate colour-coded shark numbers in *Figures 1* and *2*; DNR, did not report.

• Supplementary file 2. Testing habitat selection of blue sharks above the eastern tropical Atlantic (ETA) oxygen minimum zone (OMZ). Real blue shark movements were compared to correlated random walk models for those sharks tagged in the Azores that encountered the ETA OMZ off western Africa. Proportion test described in *Jaine et al., 2014*; ns, non-significance.

• Supplementary file 3. Fishing activity information for the different major fishing areas in the Atlantic. Shaded rows denote FAO area encompassing a permanent oxygen minimum zone.

• Transparent reporting form

### Data availability
All data needed to assess the conclusions are available in the main paper and Supporting Data. The data files are available to download from GitHub: https://github.com/GlobalSharkMovement/Blue-SharkOMZ/ (copy archived at https://archive.softwareheritage.org/swh:1:rev:b302fc919ad48114322570cae3e7dd63eaba5764/) The data files made available with the paper on GitHub are: (1) Raw summary shark dive data both inside and outside the OMZ area (shark_dive_data.csv); (2) Figure 1. Raster of DO at 100m used in shark movement/dive analysis at $0.25 \times 0.25°$ (do_av2009-2016_100m.csv); (3) Figure 6D. Raster of the shark MDD depth for present-day at $1 \times 1°$ for the area analysed (shark_MDD_prst_fao.csv); (4) Figure 7. Raster of vessel monitoring system (VMS) fishing effort data at $1 \times 1°$(VMS_f-effort.csv); (5) Raster of vessel monitoring system (VMS) fishing intensity data at $1 \times 1°$(VMS_f-intens.csv); (6) Figure 8A. Raster of Spanish longline logbook catch-per-unit-effort data at $1 \times 1°$for the area analysed (cpue_spanishLL.csv). The blue shark is listed as Near Threatened in the IUCN Red List, therefore the raw, detailed location data are considered sensitive information and, consequently, the raw tracks are not freely available so as not to encourage further fisheries interactions. Raw VMS data are owned by the Spanish and Portuguese governments and written request to them for access is usually required.

The following dataset was generated:

| Author(s) | Year | Dataset title | Dataset URL | Database and Identifier |
|---|---|---|---|---|
| Sims DW, Afonso P, Queiroz N | 2020 | Effects of ocean hypoxia and fishing on blue sharks | https://github.com/GlobalSharkMovement/BlueSharkOMZ/ | GitHub, BlueShark |

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
