## [Decision Letter]

**Acceptance summary:**

This paper combines animal-tracking technologies with environmental modelling to reveal the behavioral responses of blue sharks to low dissolved oxygen concentrations. The findings have large implications for the conservation and management of this wide-ranging species under high fishing pressure in an era of ocean de-oxygenation.

**Decision letter after peer review:**

[Editors’ note: the authors submitted for reconsideration following the decision after peer review. What follows is the decision letter after the first round of review.]

Thank you for submitting your work entitled "Climate-driven ocean deoxygenation leads to a top predator habitat compression more prone to overfishing" for consideration by *eLife*. Your article has been reviewed by three peer reviewers, including Yuuki Watanabe as the Reviewing Editor and Reviewer #1, and the evaluation has been overseen by a Senior Editor. The following individual involved in the review of your submission has agreed to reveal their identity: Nicholas Payne (Reviewer #2).

Our decision has been reached after consultation between the reviewers. Based on these discussions, and the individual reviews appended below, we regret to inform you that we cannot offer to publish your work in *eLife*.

All three reviewers acknowledge the novelty of the data and analyses presented. As you will see, however, although reviewers #1 and #2 commented very favourably, reviewer #4 raised several major concerns. During their consultation discussions, the reviewers agreed that the following points were particularly important:

– Oxygen concentration and partial pressure (PO2) must be corrected by temperature (as explained in detail by reviewer #4).

– The language is too strong in places (e.g., habitat loss due to ocean deoxygenation) and must be toned down, keeping in mind that causation has not been demonstrated and that other factors (e.g., water temperature and prey availability) may also play a role.

We believe that addressing these issues (and the many other points raised by the reviewers) will require substantial reanalysis and rewriting, and that this may fundamentally change the conclusions of the article, this is why we have decided to reject the manuscript in its present form. If you think you can fully address the points raised by the reviewers, we would be prepared to consider a resubmission. Please note, however, that resubmission does not guarantee re-review, let alone eventual acceptance, and that we are likely to consult the original reviewers again. In any case, we hope that our reviewers' comments will help you improve what could potentially be an important paper.

Reviewer #1:

The authors analyzed horizontal and vertical movements of blue sharks in relation to model-derived dissolved oxygen. They found that shark movements are affected by oxygen minimum zone and suggest that sharks will become more vulnerable to fishing activity in the future as oxygen minimum zone expands. They also analyzed fishing activity (boat movements and catches) to strengthen their argument.

Overall, I think this paper is timely and strong. I was especially impressed by fishing activity analysis, because I would consider publishing shark movement data without such analyses.

Looking at the results the effect of habitat compression is less clear than what I expect from the main story of the paper. It seems that strong effects can only be seen in the frequency of very deep dives (which occur only occasionally) and that mean swimming depth is only weakly affected. This is understandable, given that blue sharks are tolerant of wide environmental conditions (dissolved oxygen and water temperature). Although I agree with the main story of the paper, I suggest that more careful wording be used throughout the paper.

Reviewer #2:

I found this to be a great and important study that links pelagic shark distributions to one of the lesser-studied but fundamentally important oceanographic variables (oxygen availability), while also invoking fishery implications. I have a few queries below, which I don't anticipate will require major changes, and I would like to see this work published in *eLife*.

1) I agree that the correlations between depth use and oxygen availability (by the way, I appreciate presentation of BOTH mg/l and PO2) are compelling evidence that oxygen regulates the species' distribution, via physiological limitation. Nevertheless, many other parameters also change with depth (temperature is an obvious one), and are not really considered in the current manuscript. I do not consider this a major weakness of the study, but rather think a few additional sentences at key points in the discussion would help frame the findings in a way that gives the reader a better appreciation of the role of oxygen per se. For example, that PO2 provides almost the same conclusions as mg/l shows that temperature (and its influence on DO solubility) is unlikely to be the ultimate cause of the shark depth shifts. Some brief but careful text on such points ought to strengthen the manuscript even further.

2) The use of Argos data to explore selection for OMZ surface waters is interesting, but not central to the key findings, and seems complicated by the fact that only those individuals (6 of 22 tagged) that initially swam toward the OMZ were considered in analysis of selection for the OMZ. This strikes me as somewhat circular. I suggest either removing this element or adding a clearer explanation of the rationale.

3) The presentation of P_crit_ values from the literature is welcome, and provides useful context. It seems most data are derived from teleosts, which may respond to hypoxia in a different way to elasmobranchs. This point could be mentioned in the relevant passages as a way of calling for future hypoxia experiments with elasmobranchs, to move toward greater mechanistic understanding of why sharks avoid certain PO2s.

Reviewer #4:

I have reviewed this manuscript once before for a different journal. As far as I can tell there are no substantial changes from the version I saw previously.

This manuscript reports results of deployed tags on blue sharks in regions with and without pronounced oxygen minimum zones. As has been reported for many open ocean animals, from large predatory fishes and squids to zooplankton, the vertical habitat is altered by oxygen whether due to direct physiological limitations or altered predator-prey dynamics. This study is unique in attempting to model the potential increase in exposure to fishing pressure due to expanding low oxygen zones. Unfortunately, I see a few important flaws that prevent me from recommending this paper for publication.

1) The present manuscript presents oxygen concentration and partial pressure (PO2, kPa) but apparently did not calculate it correctly. The PO2 presented is directly proportional to the oxygen concentration presented, which is only true if temperature is constant…which it isn't. The manuscript needs to provide temperature profiles for the regions occupied by sharks and present oxygen partial pressure along with it. If the water is colder in OMZ regions (which it appears to be), surface waters would have higher oxygen concentrations at the same PO2 (21 kPa = air-saturation). At depth, the oxygen concentration declines more in OMZ regions, but depending on temperature profiles, the change in PO2 may be more or less significant. According to Figure 3B, sharks do not dive deep in cold water despite high oxygen concentration. So there is clearly more going on here and I don't think this paper has quite got it sorted out.

2) There is a long discussion of the oxygen concentrations that are thought to be detrimental or limiting to one species or another or that correspond to maximum dive depths. A threshold of 3.5 ml is concluded to be a likely limit for blue sharks. None of those estimates mean much without a temperature reference and they don't apply across ocean basins. Animals can be limited in depth distribution by temperature, oxygen, pressure or simply "choose" not to dive deep because they don't need to if food is abundant shallow. Adaptation to oxygen minimum zone regions will result in lower thresholds (higher tolerance). The dive depth changes observed here and in previous studies, correlate with oxygen. They may be caused by physiological limits of the sharks, but they may be due to ecological considerations that given little attention in the present study. If the dive depth changes are due to ecology, rather than physiological effects of low oxygen, modeling future habitat compression becomes a much more complicated issue. I think it’s equally likely that, rather than oxygen per se, organisms are greater abundance due to upwelling of nutrients (which also contributes to low oxygen), more food is available in surface waters so predators (including fisherman) congregate there and are not required to dive (or fish) as deep to find food. If it were the case that the sharks needed to dive deeper to find food but couldn't due to oxygen, then they would not congregate there. So this paper needs to think carefully about how causation is assigned and how that effects efforts to predict changes to depth distribution in the future.

3) If a hook is set in shallow water, and your model suggests that sharks will spend more time in shallow water, then it is obvious that your susceptibility model will show increased exposure to hooks. So I don't see how the hypothesis is directly tested here anymore than in other papers that simply stated that reduced dive depth may increase susceptibility to fishing pressure.

In summary, we simply don't know enough about blue shark (or any shark) responses to temperature and oxygen, their natural habitat ranges, their feeding preferences, and the ecological changes occurring due to climate change, to warrant the wild speculation of habitat loss in this species. This paper contains useful and interesting data on the diving depths of a handful of tagged blue sharks and shows convincingly that they dive deeper in regions with higher oxygen. However, causation has not been demonstrated and there is not sufficient data in the literature to base speculation. I would like to see this paper focus on the data and eliminate the modeling.

"There is a growing body of studies examining how horizontal and vertical distributions of marine animals including fish have become altered by climate-driven changes in oxygen, temperature and other stressors (e.g. Pörtner and Knust, 2007; Ekau et al., 2010; Deutsch et al., 2015; Mislan et al., 2017; Stortini et al., 2017). "

None of the studies cited examines how distributions have already become altered by climate as suggested by this text. All of them correlate existing habitat with oxygen/temperature but none provide evidence that habitats have already moved at all in relation to climate change.

"By contrast, very few studies have examined how pelagic shark distributions will be affected by future climate-driven changes in environmental conditions (Payne et al., 2017)."

This statement is simply not true. In fact, there are many shark studies, arguably more in proportion to their abundance and ecological importance relative to e.g. zooplankton, that are not cited here (listed below).

Estimating oxygen uptake rates to understand stress in sharks and rays

Scaling the influence of temperature and oxygen on a vertically migrating deepwater shark

Differential Effects of Temperature on Oxygen Consumption and Branchial Fluxes of Urea, Ammonia, and Water in the Dogfish Shark (Squalus acanthias suckleyi)

In situ swimming behaviors and oxygen consumption rates of juvenile lemon sharks (Negaprion brevirostris)

… of Temperature, Salinity, and Dissolved Oxygen on the Stress Response of Bull (Carcharhinus leucas) and Bonnethead (Sphyrna tiburo) Sharks after Capture and.…

Patterns of long-term climate variability and predation rates by a marine apex predator, the white shark Carcharodon carcharias

Blood O2 affinity of a large polar elasmobranch, the Greenland shark Somniosus microcephalus

Powering ocean giants: the energetics of shark and ray megafauna

Climate cooling and clade competition likely drove the decline of lamniform sharks

Examining Shortfin Mako and Blue Shark Movements in Relation to the Southern California Bight Oxygen Minimum Zone

[Editors’ note: further revisions were suggested prior to acceptance, as described below.]

Thank you for submitting your article "Climate-driven deoxygenation elevates fishing vulnerability for the ocean's widest ranging shark" for consideration by *eLife*. Your article has been reviewed by three peer reviewers, including Yuuki Watanabe as the Reviewing Editor and Reviewer #1, and the evaluation has been overseen by a Senior Editor. The following individuals involved in the review of your submission have agreed to reveal their identity: Kevin Weng (Reviewer #2); Jayson Semmens (Reviewer #3).

The reviewers have discussed their reviews with one another, and the Reviewing Editor has drafted this decision letter to help you prepare a revised submission.

The three reviewers agree that this article represents a timely analysis of pelagic fish behaviour in relation to dissolved oxygen concentrations, which will be of interest to the broad readership of *eLife*.

Summary:

The authors studied the horizontal movements and diving patterns of blue sharks over a vast area in the Atlantic, including the eastern tropical Atlantic where the oxygen minimum zone (OMZ) is known to be expanding. Specifically, they examined the associations between shark movements and model-derived oxygen concentrations in the water. They also analyzed the movements of long-line vessels and shark catch data. Taken together, they showed that sharks tend to dive more shallowly in the OMZ area, suggesting that they will be more vulnerable to long-line fisheries as OMZ expands in the future.

Revisions:

It would be helpful to briefly discuss the different kinds of oxygen measurements (respirometry-based P_crit_ from lab; tracking-based O2 that animals did NOT enter, from the field; fishery-dependent O2 levels where CPUE was low or high). This may be best accomplished near the Discussion paragraph two of subsection “Environmental drivers of shark dive depths”. It is important to note that the P_crit_ value is based upon respirometry experiments, and is not the same as the pO2 that causes a species to avoid a volume of water. The lab results show us the pO2 at which the animal can no longer sustainably maintain all function; whereas in the field, an animal may decide to enter low O2 waters for a short period of time (and incur oxygen debt).

---

## [Author Response]

[Editors’ note: the authors resubmitted a revised version of the paper for consideration. What follows is the authors’ response to the first round of review.]

Reviewer #1:The authors analyzed horizontal and vertical movements of blue sharks in relation to model-derived dissolved oxygen. They found that shark movements are affected by oxygen minimum zone and suggest that sharks will become more vulnerable to fishing activity in the future as oxygen minimum zone expands. They also analyzed fishing activity (boat movements and catches) to strengthen their argument.Overall, I think this paper is timely and strong. I was especially impressed by fishing activity analysis, because I would consider publishing shark movement data without such analyses.Looking at the results the effect of habitat compression is less clear than what I expect from the main story of the paper. It seems that strong effects can only be seen in the frequency of very deep dives (which occur only occasionally) and that mean swimming depth is only weakly affected. This is understandable, given that blue sharks are tolerant of wide environmental conditions (dissolved oxygen and water temperature). Although I agree with the main story of the paper, I suggest that more careful wording be used throughout the paper.

We agree with the reviewer that the results regarding mean depth and frequency of deep dives were not sufficiently well described so as to be as clear as possible. In the revision we have toned down the manuscript throughout by emphasising that the principal effect of environment on blue shark diving was to reduce deep dive frequency with only a relatively modest effect on percentage time spent at depth. We describe the results in more detail and discuss the main patterns in the Discussion, for example:

“Despite this, we found that the average maximum dive depth of tracked blue sharks in the ETA OMZ was 40% less than the mean depth attained outside the area, together with a greatly reduced frequency of deep diving below 600 m inside the OMZ. […] Similarly, our results show blue shark vertical movements in the OMZ area principally comprised a reduced mean maximum depth and frequency of deep dives >600 m, rather than a substantial shift in the temporal use of the uppermost water layer.”

Reviewer #2:I found this to be a great and important study that links pelagic shark distributions to one of the lesser-studied but fundamentally important oceanographic variables (oxygen availability), while also invoking fishery implications. I have a few queries below, which I don't anticipate will require major changes, and I would like to see this work published in eLife.1) I agree that the correlations between depth use and oxygen availability (by the way, I appreciate presentation of BOTH mg/l and PO2) are compelling evidence that oxygen regulates the species' distribution, via physiological limitation. Nevertheless, many other parameters also change with depth (temperature is an obvious one), and are not really considered in the current manuscript. I do not consider this a major weakness of the study, but rather think a few additional sentences at key points in the discussion would help frame the findings in a way that gives the reader a better appreciation of the role of oxygen per se. For example, that PO2 provides almost the same conclusions as mg/l shows that temperature (and its influence on DO solubility) is unlikely to be the ultimate cause of the shark depth shifts. Some brief but careful text on such points ought to strengthen the manuscript even further.

We would like to thank the reviewer for drawing attention to this important point which was not adequately described in the original manuscript. We agree that temperature should have figured more in our Results and Discussion. In the revised paper we now include expanded descriptions and new figures: (i) shark MDD depths overlaid on water temperature alongside those for DO in Figure 2, (ii) new panels in Figure 3 showing the temperature at time-at-depth and at MDD depths, (iii) temperature profiles alongside DO, PO2 and NPP in Figure 4, and (iv) a new Figure 5 and Figure 5—figure supplement 1 showing percentage time individual sharks spend at different depth, DO concentrations and water temperature.

Within the paper text we now include a new Results section “Environmental influences on diving” that specifically describes shark vertical movements in relation to low water temperatures and water column temperature gradients. We have also expanded the section “Modelling shark responses to environmental variables” to include a more detailed description of the GAMM output with respect to temperature, both surface and at depth.

In the Discussion we now include a new section on “Environmental drivers of shark dive depths” and an expanded section focused on the GAMM output “DO, temperature and NPP effects”. We also now make specific reference to temperature in the Abstract.

2) The use of Argos data to explore selection for OMZ surface waters is interesting, but not central to the key findings, and seems complicated by the fact that only those individuals (6 of 22 tagged) that initially swam toward the OMZ were considered in analysis of selection for the OMZ. This strikes me as somewhat circular. I suggest either removing this element or adding a clearer explanation of the rationale.

The additional Argos tracks were shown for context, to demonstrate that not all blue sharks move south but those that do and encounter the OMZ are more likely to show preference to remain there longer than expected compared to random models. We have clarified this rationale in the revised paper which now reads:

“The purpose of using Argos transmitters was to determine the broad extent of horizontal movements and space utilization of sharks in OMZ and adjacent areas of the North Atlantic, and as a spatial context for assessing potential habitat selection of sharks in the OMZ area, while PSAT tags were used to record spatially explicit swimming depths of sharks directly in relation to modelled DO and other environmental variables in OMZ and adjacent areas.”

3) The presentation of P_crit_ values from the literature is welcome, and provides useful context. It seems most data are derived from teleosts, which may respond to hypoxia in a different way to elasmobranchs. This point could be mentioned in the relevant passages as a way of calling for future hypoxia experiments with elasmobranchs, to move toward greater mechanistic understanding of why sharks avoid certain PO2s.

We would like to thank the reviewer for raising this important point. We have revised the paper in two passages to draw attention to the lack of P_crit_ values available for large pelagic sharks. We now clarify that only 2 shark species were included in the average P_crit_ value determined for 151 fish species by Rogers et al., 2016, and they were for benthic species:

“The critical oxygen level (*P*_crit_) – the level below which a stable rate of oxygen uptake (oxyregulation) can no longer be maintained and becomes dependent upon ambient oxygen availability (oxyconforming) – was found to average 5.15 ± 2.21 kPa for 151 fish species, ranging from 1.02 – 16.2 kPa, although only 2 species included were sharks (Rogers et al., 2016; Gallo et al., 2019). For southern Bluefin tuna, *Thunnus maccoyii*, *P*_crit_ was around 5.5 kPa at 19 °C (Rogers et al., 2016), however similar data for pelagic sharks are lacking. The mean *P*_crit_ for the bottom-dwelling catshark *Scyliorhinus canicula* was determined to be 6.5 and 8.0 kPa at 12 and 17 °C respectively (Rogers et al., 2016). For blue sharks, we estimated from modelled oxygen concentrations during vertical movements that the minimum oxygen conditions potentially encountered were 3.58 – 4.27 kPa at 12 – 18 °C, suggesting blue sharks enter waters with low partial pressures of oxygen that in many fish species, including benthic sharks, would be oxygen levels that elicit oxyconforming responses (Rogers et al., 2016).”

Furthermore, we include a sentence in the section on “Environmental drivers of shark dive depths” where we discuss the need for DO sensors on shark tags. Additionally, in The Discussion of “DO, temperature and NPP effects” we conclude the section:

“These types of sensors will enable new research on the physiological responses of ram-ventilating pelagic sharks to low DO in the wild which, when combined with novel sea-going respirometers (Payne et al., 2015) for direct determination of *P*_crit_, will pave the way for a mechanistic understanding of hypoxia tolerance in large shark species.”

Reviewer #4:I have reviewed this manuscript once before for a different journal. As far as I can tell there are no substantial changes from the version I saw previously.

We did submit a previous version of the paper to a journal, however the paper submitted to e*Life* contained substantial additions as recommended by the reviewers of the previous journal. The submission to *eLife* included the new calculations of pO2 from DO, temperature, salinity and pressure data (see below). It also included new spatially-referenced blue shark catch data obtained from the logbooks of the Spanish longlining fleet (which was also satellite tracked using VMS), neither of which were trivial to obtain. We should like to clarify that we always listen to the constructive criticisms of reviewers to improve our work and have done so not only in preparing our initial submission to e*Life* but also in this revised paper.

This manuscript reports results of deployed tags on blue sharks in regions with and without pronounced oxygen minimum zones. As has been reported for many open ocean animals, from large predatory fishes and squids to zooplankton, the vertical habitat is altered by oxygen whether due to direct physiological limitations or altered predator-prey dynamics. This study is unique in attempting to model the potential increase in exposure to fishing pressure due to expanding low oxygen zones. Unfortunately, I see a few important flaws that prevent me from recommending this paper for publication.1) The present manuscript presents oxygen concentration and partial pressure (PO2, kPa) but apparently did not calculate it correctly. The PO2 presented is directly proportional to the oxygen concentration presented, which is only true if temperature is constant…which it isn't.

We thank the reviewer for raising this point and giving us the opportunity to clarify the methods we used that were fully described in the paper. PO2 was calculated using the R package ‘*rMR*’ which considers DO, temperature, salinity and pressure to calculate PO2, which we did for each individual shark geolocation. The Materials and methods section of our manuscript and the revised paper provided these details:

“We also analysed the diving movements of blue sharks in relation to partial pressures of oxygen calculated from modelled surface and at-depth DO, temperature and salinity data and report these in the Results and Discussion and in Figure 4 and Figure 2—figure supplement 1. The partial pressure of oxygen in the water was calculated using the R library ‘*rMR*’, using temperature, salinity and DO from CMEMS global ocean biogeochemistry non-assimilative hindcast (PICES; 1998-2016) and global ocean physics reanalysis (GLORYS12V1; 1993-2016) products, interpolated to matching depth levels for both products at 0.25° spatial resolution. Spatial geolocation error of shark tracking was taken into consideration by averaging pO_2_ for 1.25° in latitude and 0.75° in longitude around each shark position.”

The Figure 2—figure supplement 1 is now included in the revised paper to make clear which data were used to calculate PO2 for each shark geolocation across individuals, and to show how the profiles of DO, temperature and salinity vary and interact at depth. For example, although near the surface PO2 and DO may seem directly proportional to one another, a closer look emphasizes the differences between the two measures. Just to clarify, in the area of the ETA OMZ, within the mixed layer depth, ocean circulation and atmospheric mixing allow for high concentrations of dissolved oxygen despite the high temperatures in shallow waters. However, this area is characterized by a current system that causes poor mid-water ventilation and strong stratification of the water profile with high productivity, leading to a large deposition of organic matter with consequent decomposition that consumes the little available oxygen, creating a mid-water OMZ. Thus, in the OMZ, despite the decreasing temperature with depth, there is not a direct increase in PO2 as the available DO is minimum. PO2 increasing in cold waters can then be observed below the OMZ, as mentioned above, where deep-water currents increase ocean ventilation and the cold waters allow for higher oxygen solubility, increasing the available PO2.

The manuscript needs to provide temperature profiles for the regions occupied by sharks and present oxygen partial pressure along with it.

This was suggested by reviewer #2 also and has been changed accordingly. In the revised paper we now include expanded descriptions and new figures: (i) shark MDD depths overlaid on water temperature alongside those for DO in Figure 2, (ii) new panels in Figure 3 showing the temperature at time-at-depth and at MDD depths, (iii) temperature profiles alongside DO, PO2 and NPP in Figure 4, and (iv) a new Figure 5 and Figure 5—figure supplement 1 showing percentage time individual sharks spend at different depth, DO concentrations and water temperature.

Within the paper text we now include a new Results section “Environmental influences on diving” that specifically describes shark vertical movements in relation to low water temperatures and water column temperature gradients. We have also expanded the section “Modelling shark responses to environmental variables” to include a more detailed description of the GAMM output with respect to DO, temperature and NPP, both surface and at depth.

In the Discussion we now include a new section on “Environmental drivers of shark dive depths” and an expanded section focused on the GAMM output “DO, temperature and NPP effects”. We also now make specific reference to temperature in the Abstract.

If the water is colder in OMZ regions (which it appears to be), surface waters would have higher oxygen concentrations at the same PO2 (21 kPa = air-saturation). At depth, the oxygen concentration declines more in OMZ regions, but depending on temperature profiles, the change in PO2 may be more or less significant. According to Figure 3B, sharks do not dive deep in cold water despite high oxygen concentration. So there is clearly more going on here and I don't think this paper has quite got it sorted out.

As mentioned in answer to the previous point, we have substantially expanded our Results and Discussion to include more detailed considerations of DO and temperature and with supporting figures. We realise that these were not as clear in the previous version as they could have been.

To clarify, the original Figure 3B showed the sea surface temperature (SST) for each maximum daily dive (MDD) depth attained along a shark track. This erroneously included positions in the western north Atlantic with cold SSTs, which confused the interpretation of SST and MDD depth in the OMZ and immediately adjacent waters. We have corrected Figure 3 to include geolocations only in the eastern Atlantic and also include two further panels (new Figure 3C, D) which clarify that the water temperatures encountered by sharks during dives to depth are not substantially different (albeit slightly lower at the same depth) at similar depths inside (low DO at depth) and outside (higher DO at depth) the OMZ. Figure 3D clarifies that sharks do dive into cold waters in high and low DO (and PO2), but in low DO waters the dives are less frequent. Throughout the revised paper we consider time-at-depth and MDD depth trends in relation to DO and temperature in much greater detail than previously.

Furthermore, we have added a new Figure 5, which shows the time that sharks spend at different depths, DO concentrations and temperatures. This clarifies that sharks spent most time in the top 100 m regardless of whether sharks are inside or outside the OMZ, and although they are diving in both habitats, the temperatures encountered at depth are similar across habitats and well within their normal preferred temperature range, whereas DO varies substantially inside the OMZ compared to outside.

We feel that this provides much more detail on sharks’ responses to DO and temperature than previously.

2) There is a long discussion of the oxygen concentrations that are thought to be detrimental or limiting to one species or another or that correspond to maximum dive depths. A threshold of 3.5 ml is concluded to be a likely limit for blue sharks. None of those estimates mean much without a temperature reference and they don't apply across ocean basins. Animals can be limited in depth distribution by temperature, oxygen, pressure or simply "choose" not to dive deep because they don't need to if food is abundant shallow. Adaptation to oxygen minimum zone regions will result in lower thresholds (higher tolerance). The dive depth changes observed here and in previous studies, correlate with oxygen. They may be caused by physiological limits of the sharks, but they may be due to ecological considerations that given little attention in the present study. If the dive depth changes are due to ecology, rather than physiological effects of low oxygen, modeling future habitat compression becomes a much more complicated issue. I think it’s equally likely that, rather than oxygen per se, organisms are greater abundance due to upwelling of nutrients (which also contributes to low oxygen), more food is available in surface waters so predators (including fisherman) congregate there and are not required to dive (or fish) as deep to find food. If it were the case that the sharks needed to dive deeper to find food but couldn't due to oxygen, then they would not congregate there. So this paper needs to think carefully about how causation is assigned and how that effects efforts to predict changes to depth distribution in the future.

We thank the reviewer for raising these important points and agree that there was insufficient coverage of alternative explanations for shoaling MDD depth in our original paper, including a more in-depth discussion of ecological considerations. We have addressed this in the revision by extending our Results and Discussion with new sections to specifically address the role of several factors in addition to DO in contributing to decreased MDD depths in the OMZ. In particular, we have expanded the section “Modelling shark responses to environmental variables” to include a clearer description of the GAM model output. We emphasise that the model shows blue shark MDD depths are driven by the combined effects of DO, SST and net primary production. This addresses directly the reviewer’s point that factors in addition to DO need to be more fully considered.

For example, the revised part of the Abstract regarding environmental effects now reads:

“Here, analysis of satellite-tracked blue sharks and environmental modelling in the eastern tropical Atlantic oxygen minimum zone (OMZ) shows shark maximum dive depths decreased due to combined effects of decreasing dissolved oxygen (DO) at depth, high sea surface temperatures, and increased surface-layer net primary production. Multiple factors associated with climate-driven deoxygenation contributed to blue shark vertical habitat compression, potentially increasing their vulnerability to surface fisheries.”

Furthermore, we have included two new Discussion sections “Environmental drivers of shark dive depths” and, in particular, “DO, temperature and NPP effects”. In these sections we give a much more balanced discussion of the combined effects. Some examples include:

“The ETA OMZ is characterised by higher SSTs than adjacent waters (e.g. Figure 2A) and, indeed, we found that MDD depth increased with increasing SST up to 24 °C, with SST >24 °C being observed along blue shark tracks across the core OMZ area that coincided with the shallowest MDD depths. The combination of high SST above the OMZ lowering oxygen solubility in water, together with the increased metabolic costs (oxygen consumption) of an ectothermic shark such as the blue shark associated with occupying elevated SSTs (Payne et al., 2015), may have acted to reduce the tolerance of blue sharks to undertake deeper dives into waters with low DO at depth.”

“Results support the hypothesis that some tracked blue sharks (10 of 16; 63%) exhibited selection of habitats above the core OMZ when encountered, which could be explained by enhanced foraging opportunities for sharks above and possibly within low DO environments of the OMZ that occurred in the uppermost 150 m layer. Blue shark feed on pelagic fish prey that may be habitat compressed into surface waters by the ETA OMZ, perhaps also due to the combined effects of low DO at depth and greater feeding opportunities in the upper layer.”

“Therefore, overall, our results support the hypothesis that expanding OMZs due to climate change (shoaling low DO; increasing SST and higher upper layer NPP) will cause habitat compression of blue sharks further into surface waters above expanding OMZs and reduce habitat volumes. The results strongly suggest that abiotic (low DO, high SST) and biotic factors (NPP, hence fish prey species) contribute to the observed decreased MDD depth of blue sharks in the ETA OMZ.”

3) If a hook is set in shallow water, and your model suggests that sharks will spend more time in shallow water, then it is obvious that your susceptibility model will show increased exposure to hooks. So I don't see how the hypothesis is directly tested here anymore than in other papers that simply stated that reduced dive depth may increase susceptibility to fishing pressure.

We agree with the reviewer that the hook encounter model is premature in the absence of a more principled modelled future projection of shark MDD depth driven by multiple biotic and abiotic factors. Therefore we have removed this model results and discussion from the paper.

In summary, we simply don't know enough about blue shark (or any shark) responses to temperature and oxygen, their natural habitat ranges, their feeding preferences, and the ecological changes occurring due to climate change, to warrant the wild speculation of habitat loss in this species. This paper contains useful and interesting data on the diving depths of a handful of tagged blue sharks and shows convincingly that they dive deeper in regions with higher oxygen. However, causation has not been demonstrated and there is not sufficient data in the literature to base speculation. I would like to see this paper focus on the data and eliminate the modeling.

We agree with the reviewer that the modelled future projections of shark MDD depths and, hence, the projected habitat loss due to habitat compression will be difficult to model accurately in the absence of a fuller understanding of shark responses to environment. Therefore, we have removed the future habitat modelling from the revised paper and have instead, as described in the previous points, focused more on shark behaviour and movements in relation to environmental variables, fishing effort and spatial catch data. We feel the revised paper is now much stronger and more inclusive of the multiple environmental drivers of shark behaviour in the OMZ.

"There is a growing body of studies examining how horizontal and vertical distributions of marine animals including fish have become altered by climate-driven changes in oxygen, temperature and other stressors (e.g. Pörtner and Knust, 2007; Ekau et al., 2010; Deutsch et al., 2015; Mislan et al., 2017; Stortini et al., 2017). "None of the studies cited examines how distributions have already become altered by climate as suggested by this text. All of them correlate existing habitat with oxygen/temperature but none provide evidence that habitats have already moved at all in relation to climate change.

We have removed this section from the revised paper as it related to modelling future shark habitat which has now been removed.

"By contrast, very few studies have examined how pelagic shark distributions will be affected by future climate-driven changes in environmental conditions (Payne et al., 2017)."This statement is simply not true. In fact, there are many shark studies, arguably more in proportion to their abundance and ecological importance relative to e.g. zooplankton, that are not cited here (listed below).Estimating oxygen uptake rates to understand stress in sharks and raysScaling the influence of temperature and oxygen on a vertically migrating deepwater sharkDifferential Effects of Temperature on Oxygen Consumption and Branchial Fluxes of Urea, Ammonia, and Water in the Dogfish Shark (Squalus acanthias suckleyi)In situ swimming behaviors and oxygen consumption rates of juvenile lemon sharks (Negaprion brevirostris)… of Temperature, Salinity, and Dissolved Oxygen on the Stress Response of Bull (Carcharhinus leucas) and Bonnethead (Sphyrna tiburo) Sharks after Capture and.…Patterns of long-term climate variability and predation rates by a marine apex predator, the white shark Carcharodon carchariasBlood O2 affinity of a large polar elasmobranch, the Greenland shark Somniosus microcephalusPowering ocean giants: the energetics of shark and ray megafaunaClimate cooling and clade competition likely drove the decline of lamniform sharksExamining Shortfin Mako and Blue Shark Movements in Relation to the Southern California Bight Oxygen Minimum Zone

We have removed the statement about how pelagic shark distributions will be affected by future climate-driven changes in environmental conditions. Because we have removed the future modelling component of the study in the revised paper, along with all the sections of text relating to it, there were no appropriate places to cite the suggested literature. We appreciate that the tenth suggestion made is highly relevant to the present study, however we note that it is a Master’s thesis and has yet to appear in the peer-reviewed literature so we have not included it at this stage.

[Editors’ note: what follows is the authors’ response to the second round of review.]

Revisions:It would be helpful to briefly discuss the different kinds of oxygen measurements (respirometry-based P_crit_ from lab; tracking-based O2 that animals did NOT enter, from the field; fishery-dependent O2 levels where CPUE was low or high). This may be best accomplished near the Discussion paragraph two of subsection “Environmental drivers of shark dive depths”. It is important to note that the P_crit_ value is based upon respirometry experiments, and is not the same as the pO2 that causes a species to avoid a volume of water. The lab results show us the pO2 at which the animal can no longer sustainably maintain all function; whereas in the field, an animal may decide to enter low O2 waters for a short period of time (and incur oxygen debt).

We thank the reviewers for suggesting this helpful clarification. We have now added the suggested descriptions of the different types of oxygen measurements to the Discussion.

The revised Discussion text now reads:

“There were similarities in the DO concentrations reducing blue shark MDD depth with those observed for other large pelagic fishes. The lower habitat boundary DO concentration we found for blue shark is consistent with studies showing depth distributions of yellowfin (Thunnus albacares) and skipjack (Katsuwonus pelamis) tunas are limited by reductions in oxygen content to only 3.5 ml O2 l-1 (Gooding et al., 1981; Cayre and Marsac, 1993; Brill et al., 1994; Lowe et al., 2000). Tracking-based or fishing depth-of-capture estimates of lower DO concentrations encountered by blue sharks do not identify strict habitat boundaries however, since sharks may choose to enter lower DO concentration waters for a short time period – a behaviour confirmed by our tracking-based results – which may incur an oxygen debt. Rather, laboratory-based experimental whole-animal measurements of the critical oxygen level (P_crit_) determine the level below which a stable rate of oxygen uptake (oxyregulation) can no longer be maintained and becomes dependent upon ambient oxygen availability (oxyconforming).”